# Stochastic $L^\natural$-convex Function Minimization

**Haixiang Zhang**
Department of Mathematics
University of California, Berkeley
Berkeley, CA 94704
`haixiang_zhang@berkeley.edu`

**Zeyu Zheng**
Department of IEOR
University of California, Berkeley
Berkeley, CA 94704
`zyzheng@berkeley.edu`

**Javad Lavaei**
Department of IEOR
University of California, Berkeley
Berkeley, CA 94704
`lavaei@berkeley.edu`

## Abstract

We study an extension of the stochastic submodular minimization problem, namely, the stochastic $L^\natural$-convex minimization problem. We develop the first polynomial-time algorithms that return a near-optimal solution with high probability. We design a novel truncation operation to further reduce the computational complexity of the proposed algorithms. When applied to a stochastic submodular function, the computational complexity of the proposed algorithms is lower than that of the existing stochastic submodular minimization algorithms. In addition, we provide a strongly polynomial approximate algorithm. The algorithm execution also does not require any prior knowledge about the objective function except the $L^\natural$-convexity. A lower bound on the computational complexity that is required to achieve a high probability error bound is also derived. Numerical experiments are implemented to demonstrate the efficiency of our theoretical findings.

## 1 Introduction

As an important branch of optimization, discrete stochastic optimization problems (Futschik & Pflug, 1995; Gutjahr & Pflug, 1996; Futschik & Pflug, 1997; Kleywegt et al., 2002; Semelhago et al., 2020) appear in a wide range of applications such as image processing (Jegelka & Bilmes, 2011), pricing optimization (Ito & Fujimaki, 2016) and simulation-based optimization (Altman et al., 2003; Freund et al., 2017). Since a discrete optimization problem is generally $\mathcal{NP}$-hard to solve, specific structures are necessary to guarantee the existence of polynomial-time algorithms. In the special case when the feasible set is $\{0, 1\}^d$ for some positive integer $d$, submodular functions are viewed as the discrete analogy of convex functions in the field of combinatorial optimization. The submodular function minimization (SFM) problem has wide applications in computer vision, economics, game theory and is well-studied in literature (Lee et al., 2015; Axelrod et al., 2020; Zhang et al., 2020). However, for certain problems, because of the stochastic structure, the exact value of the objective function is not available but only noisy samples of the objective function value are available. These scenarios, for example, include the optimization of expected performances for complex stochastic systems. In contrast to the deterministic problem, the stochastic SFM problem is less understood (Ito, 2019; Chakrabarty et al., 2017; Blais et al., 2019; Axelrod et al., 2020).

In this work, we consider the stochastic minimization problem of a class of generalized submodular functions, named the $L^\natural$-*convex functions* (Murota, 2003). More concretely, a function

35th Conference on Neural Information Processing Systems (NeurIPS 2021).

$f : \{1, \ldots, N\}^d \mapsto \mathbb{R}$ is called an $L^\natural$-convex function if it satisfies the *discrete midpoint convexity*:

$$f(x) + f(y) \geq f(\lceil (x + y)/2 \rceil) + f(\lfloor (x + y)/2 \rfloor), \quad \forall x, y \in \{1, \ldots, N\}^d,$$

where $N, d$ are positive integers and $\lceil \cdot \rceil, \lfloor \cdot \rfloor$ are the ceiling and flooring functions applied component-wisely to vectors. We note that $L^\natural$-convex functions are reduced to submodular functions if $N = 2$. As an extension to the stochastic SFM problem, the stochastic $L^\natural$-convex minimization problem also appears in a variety of decision making problems. For example, the user satisfaction function of a bike-sharing system (Freund et al., 2017) and the expected total waiting time of a service network (Altman et al., 2003) are proved to be multimodular functions, which is equivalent to an $L^\natural$-convex function under a linear transformation. Since the evaluation of both functions includes the simulation of random user behaviors, we only have access to noisy objective function value oracles and thus both problems can be formulated as stochastic $L^\natural$-convex minimization problems. We refer the readers to the appendix and Murota (2003) for more concrete examples of $L^\natural$-convex minimization problems.

The contributions of this work are two-fold. For the algorithm design, we provide the first polynomial-time algorithms to achieve high-probability bounds for the stochastic $L^\natural$-convex function minimization problem. We note that we only consider *the number of objective function value evaluations* in the computational complexity, since the function value evaluation oracle is usually the major contributor to the running time of algorithms. The first algorithm is based on the observation that the Lovász extension of an $L^\natural$-convex function transforms the original problem to an equivalent continuous convex optimization problem. Therefore, we can apply the stochastic subgradient method to solve the continuous convex problem and then round the solution to an integral solution. The idea of combining the Lovász extension and the stochastic subgradient method has already been considered in Hazan & Kale (2011); Ito (2019) for stochastic online convex optimization and stochastic submodular minimization, respectively. However, the direct application of the stochastic subgradient method will lead to a large computational complexity. We add an extra truncation step to the stochastic subgradient and the computational complexity is reduced by $O(d)$. We also implement numerical experiments to show the significance of the truncation step. In the special case when $N = 2$, our problem reduces to the stochastic SFM problem and the computational complexity of the truncated subgradient method is $O(d^2)$ smaller than the stochastic cutting-plane method in Blais et al. (2019). In addition, we propose the dimension reduction method and get a strongly polynomial approximate algorithm (see the definition in Section 4.2) with computational complexity $O(d^3 N^2 (d + \log(N)) \epsilon^{-2} \log(1/\delta))$. Finally, the uniform sampling method in Zhang et al. (2021) is extended to the multi-dimensional case and the computational complexity is independent of $N$. Table 1 summarizes the upper bounds on the computational complexities to find a solution whose objective value is at most $\epsilon$ larger than the optimum with probability at least $1 - \delta$.

On the other hand, we derive lower bounds on the computational complexity for achieving a high-probability error bound. We construct a set of $L^\natural$-convex functions whose landscapes are close to each other while their optima are different with each other. Intuitively, the algorithm requires sufficiently many evaluations to differentiate those "similar" $L^\natural$-convex functions, where the similarity between functions is characterized by the KL-divergence. More rigorously, we utilize the information inequality in Kaufmann et al. (2016) and get the lower bound $O(d\epsilon^{-2} \log(1/\delta))$ on the complexity. Recently, Ito (2019) proved a lower bound for achieving an $\epsilon$-expected error. There is a difference in terms of $d$ and $\delta$ regarding the lower bound results between ours and Ito's. We note that these two lower bounds are established under different settings, i.e., the high-probability setting and the expectation setting. Hence, a directly comparison of the two lower bounds is unavailable; see the discussion in Section 5.

## 1.1 Notation

For $N \in \mathbb{N}$, we define $[N] := \{1, 2, \ldots, N\}$. For a given set $\mathcal{S}$ and an integer $d \in \mathbb{N}$, the product set $\mathcal{S}^d$ is defined as $\{(x_1, x_2 \ldots, x_d) : x_i \in \mathcal{S}, i \in [d]\}$. For two vectors $x, y \in \mathbb{R}^d$, we use $(x \wedge y)_i := \min\{x_i, y_i\}$ and $(x \vee y)_i := \max\{x_i, y_i\}$ to denote the component-wise minimum and maximum. Similarly, the ceiling function $\lceil \cdot \rceil$ and the flooring function $\lfloor \cdot \rfloor$ round each component to an integer when applied to vectors. The failing probability of algorithms is denoted as $\delta$. The notation $f = O(g)$ (resp. $f = \Theta(g)$) means that there exist absolute constants $c_1, c_2 > 0$ such that $f \leq c_1 g$ (resp. $c_1 g \leq f \leq c_2 g$). Similarly, the notation $f = \tilde{O}(g)$ (resp. $f = \tilde{\Theta}(g)$) means that there exist absolute constants $c_1, c_2 > 0$ and constant $c_3 > 0$ independent of $\delta$ such that $f \leq c_1 g + c_3$ (resp. $c_1 g \leq f \leq c_2 g + c_3$).

Table 1: Computational complexity for stochastic $L^\natural$-convex function minimization. Constants except for $d, N, \epsilon, \delta, c$ are omitted in the $O(\cdot)$ notation. The sub-optimality gap $c$ refers to the objective function value gap between the global and the best sub-optimal solutions.

| Algorithms | Unknown Sub-optimality Gap | Known Sub-optimality Gap $c$ |
|---|---|---|
| **Truncated Subgradient** | $O(d^2 N^2 \epsilon^{-2} \log(1/\delta))$ | $O(d^2 \log(N) c^{-2} \log(1/\delta))$ |
| **Multi-dim Uniform Sampling** | $O(C^d \epsilon^{-2} \log(1/\delta))$ | $O(C^d c^{-2} \log(1/\delta))$ |
| **Stochastic Cutting-plane** | $O(d^3 N^2 (d + \log(N)) \\ \epsilon^{-2} \log(1/\delta))$ | $O(d^3 \log(N)(d + \log(N)) \\ c^{-2} \log(1/\delta))$ |

## 2 Problem Formulation

We have in mind a complex stochastic system and consider the following discrete stochastic optimization problem

$$\min_x \; f(x) := \mathbb{E}[F(x, \xi_x)] \quad \text{s.t. } x \in \mathcal{X}, \tag{1}$$

in which $\mathcal{X}$ is the discrete feasible set, $\xi_x$ summarizes all the randomness supported on some proper probability space $(\mathsf{Y}, \mathcal{B}_\mathsf{Y})$ and $F : \mathcal{X} \times \mathsf{Y} \mapsto \mathbb{R}$ is a deterministic function. Specifically, $\xi_x$ summarizes all the random variables and stochastic processes involved in the system for the decision variable $x$. The function $F$ maps the decision and associated randomness into a random system performance. For example, in service systems such as the bike-sharing service, this function $F$ captures the full operational logic regarding how services are arranged. We write $\xi_{x,1}, \xi_{x,2}, \ldots, \xi_{x,n}$ as independent and identically distributed (i.i.d.) copies of $\xi_x$. We use $\hat{F}_n(x) := \frac{1}{n} \sum_{j=1}^n F(x, \xi_{x,j})$ to denote the empirical mean of the $n$ independent evaluations for the choice of decision variables $x$. The triad of the set $\mathcal{X}$, the space of randomness $(\mathsf{Y}, \mathcal{B}_\mathsf{Y})$ and the function $F(\cdot, \cdot)$ is called the *model* of problem (1). In this work, we assume the $L^\natural$-convexity of the objective function. For the ease of presentation, we assume the feasible set $\mathcal{X} = [N]^d$, where $d$ denotes the dimension and $N$ represents the number of choices on each dimension for the decision variable $x$.

**Assumption 1.** The objective function $f(\cdot)$ is an $L^\natural$-convex function on $\mathcal{X} = [N]^d$, i.e.,

$$f(x) + f(y) \geq f(\lceil (x+y)/2 \rceil) + f(\lfloor (x+y)/2 \rfloor), \quad \forall x, y \in \mathcal{X},$$

*Remark* 1. In general, the definition of $L^\natural$-convexity also requires the feasible set $\mathcal{X}$ to be an $L^\natural$-convex set, namely, the indicator function of $\mathcal{X}$ should be $L^\natural$-convex on $\mathbb{Z}^d$. In our case, the feasible set $\mathcal{X} = [N]^d$ is an $L^\natural$-convex set. Hence, we only need the discrete mid-point convexity property to define $L^\natural$-convex functions on $\mathcal{X}$.

Moreover, we assume that the distribution of the stochastic output $F(x, \xi_x)$ is sub-Gaussian.

**Assumption 2.** The distribution of $F(x, \cdot)$ is sub-Gaussian with a *known* parameter $\sigma^2$ for all $x \in \mathcal{X}$.

The knowledge of $\sigma^2$ represents the maximum level of stochasticity involved in the system performance. For scenarios where this $\sigma^2$ is not known, it has to be adaptively learned and adds additional complexity to the problem, but is not the focus of this work. We further note that our algorithm design and analysis can be generalized to any probability distribution with provable concentration inequalities. The set of all models satisfying both Assumption 1 and Assumption 2 is denoted as $\mathcal{MC}(\mathcal{X})$, or simply $\mathcal{MC}$. The main goal is to design algorithms that guarantee the selection of a near-optimal solution with high probability. Formally, this criterion is called the $(\epsilon, \delta)$-*Probably Approximately Correct (PAC)* guarantee; see Even-Dar et al. (2002); Kaufmann et al. (2016); Ma & Henderson (2017). We say that a solution $\hat{x}$, as the output of an algorithm, satisfies the $(\epsilon, \delta)$-PAC guarantee if

$$\mathbb{P}\left[ f(\hat{x}) - \min_{x \in \mathcal{X}} f(x) \geq \epsilon \right] \leq \delta,$$

where the probability is calculated over the distribution of $\hat{x}$.

An algorithm is called an $[(\epsilon, \delta)\text{-PAC}, \mathcal{MC}]$-algorithm if it always returns an $(\epsilon, \delta)$-PAC solution for any model in $\mathcal{MC}$. The *computational complexity* of an algorithm is defined as the number of evaluations of $F(\cdot, \cdot)$ before termination. The difficulty of satisfying the PAC guarantee for any model in $\mathcal{MC}$ is characterized by the worst-case performance of an optimal algorithm.

**Definition 1.** The **computational complexity** of the $(\epsilon, \delta)$-PAC guarantee and class $\mathcal{MC}$ is defined as

$$T(\epsilon, \delta, \mathcal{MC}) := \inf_{\mathbf{A} \text{ is } ((\epsilon, \delta)\text{-PAC}, \mathcal{MC})} \sup_{M \in \mathcal{MC}} \mathbb{E}\left[\tau_{\mathbf{A}}\right],$$

where the random variable $\tau_{\mathbf{A}}$ is the computational complexity for each implementation of $\mathbf{A}$.

The reminder of this paper is focused on providing upper bounds on $T(\epsilon, \delta, \mathcal{MC})$ by designing algorithms and providing lower bounds by an information-theoretical analysis. In this paper, we consider the behavior of algorithms in the high-probability regime, namely, in the setting when the failing probability $\delta$ is small enough.

**Assumption 3.** The failing probability $\delta$ is sufficiently small.

In general, the sub-Gaussian randomness leads to

$$T(\epsilon, \delta, \mathcal{M}) = C(\epsilon, \mathcal{M}) \cdot \log\left(1/\delta\right) \quad 0 < \delta \ll 1,$$

where $C(\epsilon, \mathcal{M})$ is some function only depending on the precision $\epsilon$ and the model class $\mathcal{M}$. For example, it is known that the sample complexity is $\Theta(N^d \epsilon^{-2}) \log(1/\delta)$ for the general best-arm selection problem (Even-Dar et al., 2002) when no convex structure is in presence.

## 3 Truncated Stochastic Subgradient Algorithms

In this section, we develop gradient-based algorithms for problem (1). We first show that the properties of $L^\natural$-convex functions make it possible to construct a continuous convex function on the continuous cube $[1, N]^d$ that has the same set of optima as the original objective function. Then, a rounding process is designed to round the solution of the continuous problem to an integral solution. Finally, the truncated stochastic subgradient algorithm is proposed and its computational complexity is analyzed. Moreover, in this section, we make the assumption that the $\ell_\infty$-Lipschitz constant of the objective function can be estimated a priori. For example, this estimation is possible for service systems optimization, where it represents an upper bound on how much impact an additional server over a certain time range contributes to the system performance.

**Assumption 4.** The $\ell_\infty$-Lipschitz constant of $f(x)$ is known to be $L$ a priori. Namely, we know beforehand that

$$|f(x) - f(y)| \leq L, \quad \forall x, y \in \mathcal{X} \quad \text{s.t. } \|x - y\|_\infty \leq 1.$$

### 3.1 Reduction to a continuous convex optimization problem

We first show that the original discrete minimization problem (1) can be equivalently transformed to a continuous convex optimization problem. Based on the fact that the restriction of an $L^\natural$-convex function to a hypercube is a submodular function (Murota, 2003), we show that we can piece together the Lovász extension of the objective function in each hypercube to form a continuous convex function on $\text{conv}(\mathcal{X}) = [1, N]^d$. More importantly, it can be shown that the continuous convex function has exactly the same global minima as the original $L^\natural$-convex function and an unbiased subgradient estimator is available at each point at the cost of $2d$ evaluations to $F(\cdot, \cdot)$.

**Theorem 1.** *Suppose that function $f(\cdot)$ is an $L^\natural$-convex function on $[N]^d$. Then, there exists a continuous convex function $\tilde{f}(\cdot)$ on $[1, N]^d$, which we also name as the Lovász extension, such that*

- *It holds $\tilde{f}(x) = f(x)$ for any $x \in \mathcal{X}$.*

- *The minimizers of $\tilde{f}$ satisfy $\arg\min_{y \in [1,N]^d} \tilde{f}(y) = \arg\min_{y \in [N]^d} f(y)$.*

- *For any point $x \in [1, N]^d$, there exists an unbiased estimator $\hat{g}_x$ to a subgradient of $\tilde{f}(\cdot)$ at point $x$, which can be evaluated via $2d$ evaluations to $F(\cdot, \cdot)$.*

---

**Algorithm 1** Rounding process to a feasible solution

---

**Input:** Model $\mathcal{X}, \mathcal{B}_Y, F(x, \xi_x)$, optimality guarantee parameters $\epsilon, \delta, (\epsilon/2, \delta/2)$-PAC solution $\bar{x}$ to problem (2).
**Output:** An $(\epsilon, \delta)$-PAC solution $x^*$ to problem (1).
 1: Compute the neighbouring points (defined in the appendix) of $\bar{x}$, denoted as $S^0, \dots, S^d$.
 2: Simulate at $S^i$ until the $1 - \delta/(4d)$ confidence half-width of $\hat{F}_n(S^i)$ is smaller than $\epsilon/4$ for all $i$.
 3: Return the optimum $x^* \leftarrow \arg\min_{S \in \{S^0, \dots, S^d\}} \hat{F}_n(S)$.

---

---

**Algorithm 2** Projected and truncated SSGD method for the PAC guarantee

---

**Input:** Model $\mathcal{X}, \mathcal{B}_Y, F(x, \xi_x)$, optimality guarantee parameters $\epsilon, \delta$, number of iterations $T$, step size $\eta$, truncation threshold $M$.
**Output:** An $(\epsilon, \delta)$-PAC solution $x^*$ to problem (1).
 1: Choose the initial point $x^0 \leftarrow ((N+1)/2, \dots, (N+1)/2)$.
 2: **for** $t = 0, \dots, T-1$ **do**
 3:     Generate a stochastic subgradient $\hat{g}^t$ at $x^t$.
 4:     Truncate the stochastic subgradient $\tilde{g}^t \leftarrow \mathcal{T}_M(\hat{g}^t)$.
 5:     Update $x^{t+1} \leftarrow \mathcal{P}(x^t - \eta\tilde{g}^t)$.
 6: **end for**
 7: Compute the averaging point $\bar{x} \leftarrow \left( \sum_{t=0}^{T-1} x^t \right) / T$.
 8: Round $\bar{x}$ to an integral point by Algorithm 1.

---

The proof along with a detailed discussion on $L^\natural$-convex functions is provided in the appendix.

Next, we describe the rounding process that takes an approximate solution to the relaxed problem

$$\min_{x \in [1,N]^d} \tilde{f}(x) \tag{2}$$

to an approximate integral solution to the original problem (1). We list the pseudo-code in Algorithm 1. The following theorem proves the correctness of Algorithm 1 and estimates the computational complexity. Note that all upper bounds on computational complexity in this paper are proved to hold almost surely and thus also in expectation.

**Theorem 2.** *Suppose that Assumptions 1-4 hold. The solution returned by Algorithm 1 satisfies the $(\epsilon, \delta)$-PAC guarantee. The computational complexity of Algorithm 1 is at most*

$$O\left[ \frac{d}{\epsilon^2} \log\left( \frac{d}{\delta} \right) + d \right] = \tilde{O}\left[ \frac{d}{\epsilon^2} \log\left( \frac{1}{\delta} \right) \right].$$

Although we transform the original problem to a continuous convex problem, the algorithm design in this work relies heavily on other two important properties of $L^\natural$-convex functions. The first property is that the global minima are integral points, which makes it possible to design strongly polynomial algorithms; see Section 4.2. The second property is that the Lovász extension is a piecewise linear function. When the sub-optimality gap is known, we prove that the Lovász extension satisfies the Weak Sharp Minima property and the subgradient descent method can be accelerated to achieve a better dependence on the scale $N$; see the appendix.

### 3.2 Algorithm and complexity analysis

Utilizing the tools derived in the last subsection, we can first apply the stochastic subgradient (SSGD) method to find an approximate solution to problem (2) and then round the solution to get an approximate solution to problem (1). The main difficulty of giving sharp upper bounds lies in the fact that the Lovász extension is neither smooth nor strongly-convex. This property of the Lovász extension prohibits the application of Nesterov's acceleration method (Nesterov, 1983) and variance reduction methods (Johnson & Zhang, 2013; Defazio et al., 2014).

Now, we propose the projected and truncated SSGD method for the $(\epsilon, \delta)$-PAC guarantee. The orthogonal projection onto the feasible set, which is defined as

$$\mathcal{P}(x) := (x \wedge N\mathbf{1}) \vee \mathbf{1}, \quad \forall x \in \mathbb{R}^d,$$

is applied after each iteration to ensure the feasibility of iteration point. In addition to the projection, the component-wise truncation of stochastic subgradient is critical in reducing expected computational complexities. The truncation operator with threshold $M > 0$ is defined as

$$\mathcal{T}_M(g) := (g \wedge M\mathbf{1}) \vee (-M\mathbf{1}), \quad \forall g \in \mathbb{R}^d.$$

The pseudo-code of projected and truncated SSGD method is listed in Algorithm 2. With a suitable choice of the step size, the truncation threshold and the number of iterations, Algorithm 2 returns an $(\epsilon, \delta)$-PAC solution and the computational complexity scales as $O(d^2 N^2)$.

**Theorem 3.** *Suppose that Assumptions 1-4 hold. If we choose*

$$T = \tilde{\Theta}\left[\frac{dN^2}{\epsilon^2}\log(\frac{1}{\delta})\right], \quad M = \tilde{\Theta}\left[\sqrt{\log(\frac{dNT}{\epsilon})}\right], \quad \eta = \frac{N}{M\sqrt{T}},$$

*then Algorithm 2 returns an $(\epsilon, \delta)$-PAC solution. Furthermore, we have*

$$T(\epsilon, \delta, \mathcal{MC}) = O\left[\frac{d^2 N^2}{\epsilon^2}\log(\frac{1}{\delta}) + \frac{d^3 N^2}{\epsilon^2}\log\left(\frac{d^2 N}{\epsilon^3}\right) + \frac{d^3 N^2 L^2}{\epsilon^2}\right] = \tilde{O}\left[\frac{d^2 N^2}{\epsilon^2}\log(\frac{1}{\delta})\right].$$

The intuition behind the truncation operation is that the truncation makes the norm of the subgradient smaller, which in turn allows the choice of a larger step size. With the larger step size, the convergence rate can be improved. Using the truncation operation to accelerate the algorithms is novel in the context of the stochastic $L^\natural$-convex function minimization problem. The truncation operation has been considered in other contexts, such as the sparse phase retrieval problem (Wang et al., 2017). We can prove that the computational complexity without the truncation step (i.e., $M = \infty$) is

$$\tilde{O}\left[\frac{d^3 N^2}{\epsilon^2}\log(\frac{1}{\delta})\right].$$

Hence, the truncation of stochastic subgradient is necessary for reducing the asymptotic expected computational complexity. This relation has also been observed in our numerical experiments. While the error of the normal SSGD method only contains the optimization residual and the variance terms, the error of the truncated SSGD method has an extra bias term. We note that the bias term can be made arbitrarily small with high probability by choosing large enough $M$ and utilizing the tail bound for sub-Gaussian random variables, and therefore the total error can be controlled similarly as the normal SSGD method. We mention that Algorithm 2 also applies when $\mathcal{X}$ is a general $L^\natural$-convex set.

Next, we consider a special case with additional condition that the norm of stochastic subgradients is bounded almost surely. We then show that this additional condition, whenever available, can contribute to further complexity reduction of our proposed algorithm.

**Assumption 5.** There exist a constant $G$ and an unbiased subgradient estimator $\hat{g}$ such that

$$\mathbb{P}\left[\|\hat{g}\|_1 \leq G\right] = 1.$$

Moreover, the cost of generating each $\hat{g}$ is at most $\beta$ evaluations of $F(\cdot, \cdot)$.

We note that $G$ and $\beta$ may depend on $d$ and $N$. This assumption is common when analyzing the high-probability error bound of stochastic subgradient methods (Hazan & Kale, 2011; Xu et al., 2016). We first give two examples where Assumption 5 holds.

**Example 1.** Suppose that the randomness of each feasible $x \in \mathcal{X}$ shares the same measure space, i.e., there exists a measure space $(\mathrm{Z}, \mathcal{B}_\mathrm{Z})$ such that $\xi_x$ can be any element in the measure space for all $x \in \mathcal{X}$. Moreover, for any fixed $\xi \in \mathcal{B}_\mathrm{Z}$, the function $F(\cdot, \xi)$ is also $L^\natural$-convex and has $\ell_\infty$-Lipschitz constant $\tilde{L}$. Then, the properties of $L^\natural$-convex functions imply that there exists an unbiased subgradient estimator $\hat{g}$ such that the cost of evaluating $\hat{g}$ is $d + 1$ and $\|\hat{g}\|_1 \leq 3\tilde{L}/2$. Therefore, the Assumption 5 holds with $G = 3\tilde{L}/2$ and $\beta = d + 1$. We note that this example is similar to the case when $k = d$ in Ito (2019) while the noise of our problem can be unbounded.

When the distribution at each feasible point is bounded, we show that Assumption 5 also holds.

**Example 2.** Suppose that the distribution at each point $x \in \mathcal{X}$ is bounded, namely, there exists a constant $C \geq 0$ such that

$$\mathbb{P}_{\xi_x}\left[|F(x, \xi_x)| \leq C\right] = 1, \quad \forall x \in \mathcal{X}.$$

We note that bounded distributions are sub-Gaussian with parameter $2C^2$. In this case, there exists an unbiased subgradient estimator such that Assumption 5 holds with $G = 2dC$ and $\beta = 2$.

More details about these two examples are provided in the appendix. Since the stochastic subgradient is bounded, the truncation step is unnecessary in Algorithm 2. The computational complexity of Algorithm 2 is estimated in the following theorem. The proof is similar to Lemma 10 in Hazan & Kale (2011) and, since the feasible set is the hypercube $[1, N]^d$, we replace the $\ell_2$-norm with the $\ell_\infty$-norm to bound distances between points.

**Theorem 4.** *Suppose that Assumptions 1-5 hold. If we skip the truncation step in Algorithm 2 (i.e., set $M = \infty$) and choose*

$$T = \tilde{\Theta}\left[\frac{N^2(L+G)^2}{\epsilon^2}\log(\frac{1}{\delta})\right], \quad \eta = \sqrt{\frac{dN^2}{TG^2}},$$

*then Algorithm 2 returns an $(\epsilon, \delta)$-PAC solution. Furthermore, we have*

$$T(\epsilon, \delta, \mathcal{MC}) = O\left[\frac{\beta N^2(L+G)^2 + d}{\epsilon^2}\log(\frac{1}{\delta}) + \frac{d^2G^2}{\epsilon^2}\right]$$

$$= \tilde{O}\left[\frac{\beta N^2(L+G)^2 + d}{\epsilon^2}\log(\frac{1}{\delta})\right].$$

In the case of Example 1, we have $\beta = d+1$, $G = 3\tilde{L}/2$ and then the computational complexity of Algorithm 2 is at most $\tilde{O}[dN^2(L+\tilde{L})^2\epsilon^{-2}\log(1/\delta)]$. If both Lipschitz constants are independent of $d$ and $N$, the computational complexity becomes $\tilde{O}[dN^2\epsilon^{-2}\log(1/\delta)]$, which is $O(d)$ better than the general case without Assumption 5. Moreover, in the case of Example 2, we have $G = d$ and $\beta = 2$. Hence, the computational complexity is at most $\tilde{O}[d^2N^2\epsilon^{-2}\log(1/\delta)]$, even without the truncation.

Finally, we consider the case when the global optimum is unique and the sub-optimality gap is known to be at least $c > 0$. In this case, the Lovász extension satisfies the *weak sharp minimum* condition (Burke & Ferris, 1993) and we can design an acceleration scheme similar to that in Xu et al. (2016). The acceleration scheme reduces the dependence on $N$ from $N^2$ to $\log(N)$ and we provide detailed discussions in the appendix.

## 4    Stochastic Localization Algorithms

Although the algorithms developed in Section 3 are guaranteed to find an approximate solution in polynomial time, gradient-based algorithms require the prior knowledge about the Lipschitz constant of the objective function to choose the optimal step size. In addition, the computational complexity of subgradient methods has a quadratic dependence on the Lipschitz constant. However, for certain applications, the upper bound on the Lipschitz constant may be difficult to estimate a priori or is very large so that the computational complexity is prohibitive for large-scale problems. To deal with those difficult scenarios, we utilize the discrete natural of problem (1) to design algorithms that do not require the prior knowledge about the objective function except the $L^\natural$-convexity and whose computational complexity has an upper bound that is independent of the Lipschitz constant. More specifically, we propose two approaches to deal with this problem. The first approach is to generalize the uniform sampling algorithm in Zhang et al. (2021) to the multi-dimensional case. In the second approach, we propose a strongly polynomial approximate algorithm by gradually reducing the dimension.

### 4.1    Multi-dimensional uniform sampling algorithm

We first generalize the uniform sampling algorithm in Zhang et al. (2021) to the multi-dimensional case. The basic idea is that if the computational complexity for finding an $(\epsilon, \delta)$-PAC solution has the form $C\epsilon^{-2}\log(1/\delta)$, we can view the algorithm as a (possibly biased) estimator with sub-Gaussian tails to the optimal value, where $C > 0$ is a constant. For example, the uniform sampling algorithm provides a sub-Gaussian estimator for the optimal value of problem (1) in the one-dimensional case. Now suppose that we have derived a sub-Gaussian estimator for the optimal value for the $(d-1)$-dimensional case. Consider the function

$$f^1(x_1) := \min_{x_2,\ldots,x_d \in [N]} f(x_1, \ldots, x_d), \quad \forall x_1 \in [N], \tag{3}$$

which is an one-dimensional convex function. Since the $(d-1)$-dimensional algorithm has a sub-Gaussian tail, we can view each implementation of the algorithm as a noisy function value oracle to $f^1(\cdot)$ and the uniform sampling algorithm can be modified to solve the convex problem (3). By a similar method as Zhang et al. (2021), we can prove that the computational complexity of the resulting $d$-dimensional algorithm is a constant multiple of that of the $(d-1)$-dimensional algorithm. By the induction method, we get an approximate algorithm for the multi-dimensional problem (1) whose computational complexity is independent of the Lipschitz constant $L$ and the scale $N$ in the high-probability regime. A rigorous proof to the following result is provided in the appendix.

**Theorem 5.** *Suppose that Assumptions 1-3 hold. There exists an $[(\epsilon, \delta)\text{-PAC}, \mathcal{MC}]$-algorithm with computational complexity*

$$T(\epsilon, \delta, \mathcal{MC}) = \tilde{O}\left[\frac{C^d}{\epsilon^2}\log(\frac{1}{\delta})\right],$$

*where $C$ is an absolute constant.*

The exponential dependence on the dimension $d$ makes it impossible to use the generalized algorithm even for moderate dimension problems. However, this result implies that the computational complexity almost does not grow with $N$ in the regime $N \gg d$ in the asymptotic regime with the failing probability $\delta$ is very small. Therefore this algorithm can be preferable when $d$ is small and $N$ is large.

### 4.2 Dimension reduction algorithm

Next, we take an alternative approach and derive a strongly polynomial approximate algorithm for problem (1). In this work, we do not treat the parameters $d$ and $N$ as input parameters. The message we hope to convey is that the sample complexities are upper bounded by a constant independent of the objective function. This property is important for some applications of the stochastic $L^\natural$-function minimization problem where the objective function has unfavorable properties (e.g. large Lipschitz constant, small sub-optimality gap). Such unfavorable properties may not be known a priori before the problem is solved. If the parameters $d$ and $N$ is treated as an input parameter, the proposed algorithms are pseudo-polynomial. We observe that a subgradient of a convex function provides a separation oracle to the global minima, which can be used to gradually shrink the set that contains global minima. This idea leads to the deterministic cutting-plane methods (Vaidya, 1996; Bertsimas & Vempala, 2004; Lee et al., 2015). In this work, we develop a framework for extending cutting-plane methods to the stochastic case. We prove that if we average $\tilde{O}[dN^2\epsilon^{-2}\log(1/\delta)]$ stochastic subgradients, we can get a separation oracle for the set of $\epsilon$-approximate solutions with probability at least $1 - \delta$. Utilizing this stochastic separation oracle, we may extend the deterministic cutting-plane methods to the stochastic case to find PAC solutions.

However, the stochastic cutting-plane methods still require the knowledge of the Lipschitz constant of the objective function. To design a strongly polynomial algorithm, we observe that if a convex body $P \subset \mathbb{R}^d$ has a volume smaller than $(d!)^{-1} = O[\exp(-d\log(d))]$, then all integral points in $P$ must lie on a hyperplane. Otherwise, if there exist $d + 1$ points $x_0, \ldots, x_d \in P$ that are not on the same hyperplane, then the convex body $P$ contains the polytope $\text{conv}\{x_0, \ldots, x_d\}$, which has the volume

$$\frac{1}{d!}\left|\det(x_1 - x_0, \ldots, x_d - x_0)\right| \geq \frac{1}{d!} > \text{vol}(P),$$

where $\text{vol}(P)$ is the volume of the polytope $P$. This leads to a contradiction. Hence, we may use the stochastic version of Vaidya's method (Vaidya, 1996) or the random walk-based (Bertsimas & Vempala, 2004) cutting-plane method to reduce the volume of the search polytope $P$ to $O[\exp(-d\log(d))]$ and then reduce the problem dimension by projecting the polytope onto the hyperplane. After $d - 1$ dimension reductions, we have an one-dimensional convex problem and algorithms in Zhang et al. (2021) can be applied.

**Theorem 6.** *Suppose that Assumptions 1-3 hold. Then, the dimension reduction algorithm is an $[(\epsilon, \delta)\text{-PAC}, \mathcal{MC}]$-algorithm with computational complexity*

$$T(\epsilon, \delta, \mathcal{MC}) = \tilde{O}\left[\frac{d^3 N^2(d + \log(N))}{\epsilon^2}\log(\frac{1}{\delta})\right].$$

The dimension reduction method is the first algorithm for the stochastic $L^\natural$-convex function minimization that is strongly polynomial and does not require prior knowledge about the objective function

except the $L^\natural$-convexity. Recently, Jiang (2020) proved that the LLL algorithm (Lenstra et al., 1982) is able to reduce the number of arithmetic operations to be polynomial and make the algorithm more practical, but the LLL algorithm will not reduce the number of evaluations to the stochastic objective values. Their results can be naturally extended to the stochastic case to achieve the PAC guarantee. We note that if we allow exponentially many arithmetic operations, the LLL algorithm is not necessary and the computational complexity can be reduced to $\tilde{O}[d^4 N^2 \epsilon^{-2} \log(1/\delta)]$. Similar to the gradient-based method, the acceleration scheme can reduce the number of required objective function value evaluations if we have a positive lower bound to the sub-optimality gap.

## 5  Lower Bounds

In this section, we provide a lower bound result on $[(\epsilon, \delta)\text{-PAC}, \mathcal{MC}]$-algorithms. This result shows the limit of algorithms that achieve the PAC guarantee for any stochastic $L^\natural$-convex functions. To prove lower bounds, we make non-trivial extensions to the construction in Graur et al. (2020) to get several convex models that are "similar" to each other but they have distinct optimal solutions, where the similarity between two models is characterized by the KL-divergence between their distributions. Hence, algorithms need a large number of evaluations to differentiate these models. More rigorously, the information-theoretical inequality in Kaufmann et al. (2016) is utilized to get the following lower bound:

**Theorem 7.** *Suppose Assumptions 1-3 hold. We have*

$$T(\epsilon, \delta, \mathcal{MC}) \geq \Theta\left[\frac{d}{\epsilon^2} \log(\frac{1}{\delta})\right].$$

Comparing with the lower bound for the expected error in Ito (2019), our high-probability lower bound is $O(d)$ smaller but includes an extra term $\log(1/\delta)$. Since the failing probability $\delta$ can be sufficiently small, the two lower bounds are not directly comparable. The results in Sections 3-4 imply that

$$T(\epsilon, \delta, \mathcal{MC}) = \tilde{O}\left[\frac{\min\{d^2 N^2, M^d\}}{\epsilon^2} \log(\frac{1}{\delta})\right].$$

Therefore, we conjecture that the optimal computational complexity is asymptotically independent of $N$ and leave the design of optimal algorithms for future works.

## 6  Numerical Experiments

In this section, we present a brief account of numerical experiments to illustrate several of our theoretical findings. First, we show that the extra new truncation step in the gradient-based method can achieve a better performance, compared to the gradient-based method without the truncation step. We implement Algorithm 2 with and without the truncation operation on a staffing allocation stochastic optimization problem with staffing cost constraint, which can be formulated into the regularized and the constrained forms. Details about the problem setup are provided in the appendix. We summarize the computational complexities and the objective values in Table 2. We can see that the two algorithms achieve similar objective values but the "Truncated" version observes an advantage in the computational cost compared to the "Not Truncated" version. We note that our algorithms do not require the feasible region to be a hypercube and in fact the problems considered in this section have more general feasible set with constraints. Especially, we can see that the truncation plays an important role in reducing the computational complexity, especially when the dimension is high.

Next, we compare the performances of the gradient-based methods and stochastic localization algorithms on multi-dimensional problems. To facilitate the comparison, we construct a separable stochastic convex function, which is an $L^\natural$-convex function by the results in Murota (2003). We have observed that using $(N\epsilon/4, \delta/4)$-$\mathcal{SO}$ oracles is enough for producing high-probability guarantees using the dimension reduction method within the range $1 \leq N \leq 150$. More details about the experiment setup are provided in the appendix. We summarize the results in Table 3, where the coverage rate refers to the percentage of implementations that produce an $\epsilon$-optimal solution. Since the coverage rates of the algorithms are $100\%$, the PAC guarantee is satisfied by all of the algorithms except the $(d, N) = (20, 50)$ case for the dimension reduction method. This failure is because of

Table 2: Complexity and returned objective value of Algorithm 2 on the optimal allocation problem. The cost refers to the average computational complexity in all the following tables.

| Params. | | Regularized | | | | Constrained | | | |
| --- | --- | --- | --- | --- | --- | --- | --- | --- | --- |
| | | Truncated | | Not truncated | | Truncated | | Not truncated | |
| d | N | Cost | Obj. | Cost | Obj. | Cost | Obj. | Cost | Obj. |
| 4 | 10 | 2.99e5 | 2.10e2 | 6.56e5 | 2.11e2 | 3.00e5 | 4.76e1 | 4.99e5 | 4.97e1 |
| 4 | 20 | 1.21e5 | 3.53e2 | 2.61e5 | 3.53e2 | 1.14e5 | 5.23e1 | 1.77e5 | 5.38e1 |
| 4 | 30 | 8.85e4 | 4.75e2 | 1.68e5 | 4.76e2 | 7.38e4 | 5.24e1 | 1.23e5 | 5.21e1 |
| 4 | 40 | 6.25e4 | 5.91e2 | 1.34e5 | 6.07e2 | 5.28e4 | 5.31e1 | 9.24e4 | 5.28e1 |
| 4 | 50 | 5.34e4 | 7.07e2 | 1.08e5 | 7.07e2 | 4.66e4 | 5.64e1 | 6.61e4 | 5.51e1 |
| 8 | 10 | 1.19e6 | 1.75e2 | 3.80e6 | 1.76e2 | 1.20e6 | 3.11e1 | 2.23e6 | 3.02e1 |
| 12 | 10 | 2.68e6 | 1.59e2 | 9.48e6 | 1.59e2 | 2.69e6 | 1.87e1 | 5.36e6 | 1.86e1 |
| 16 | 10 | 6.35e6 | 1.49e2 | 1.31e7 | 1.50e2 | 4.78e6 | 1.49e1 | 1.08e7 | 1.41e1 |
| 20 | 10 | 9.91e6 | 1.43e2 | 2.09e7 | 1.46e2 | 9.43e6 | 1.17e1 | 1.70e7 | 1.28e1 |
| 24 | 10 | 1.50e7 | 1.35e2 | 3.09e7 | 1.41e2 | 1.36e7 | 9.43e0 | 2.10e7 | 1.17e1 |

Table 3: Complexity and coverage rate of different algorithms on separable convex functions.

| Params. | | Truncated subgradient | | Dimension reduction | |
| --- | --- | --- | --- | --- | --- |
| d | N | Cost($10^7$) | Rate(%) | Cost($10^7$) | Rate(%) |
| 2 | 50 | 0.39 | 100.0 | 0.19 | 100.0 |
| 2 | 100 | 0.44 | 100.0 | 0.22 | 100.0 |
| 2 | 150 | 0.51 | 100.0 | 0.25 | 100.0 |
| 6 | 50 | 0.96 | 100.0 | 1.51 | 100.0 |
| 6 | 100 | 1.07 | 100.0 | 1.76 | 100.0 |
| 6 | 150 | 1.24 | 100.0 | 1.92 | 100.0 |
| 10 | 50 | 1.26 | 100.0 | 3.47 | 100.0 |
| 10 | 100 | 1.43 | 100.0 | 4.02 | 100.0 |
| 10 | 150 | 1.64 | 100.0 | 4.33 | 100.0 |
| 15 | 50 | 1.52 | 100.0 | 4.14 | 100.0 |
| 20 | 50 | 1.69 | 100.0 | 4.28 | 99.00 |

our loosened choice of the separation oracles. When the dimension $d$ is not large, the dimension reduction method has the best performance and has the advantage of not requiring any knowledge about the Lipschitz constant; when the dimension $d$ is large, the higher order dependence on $d$ makes the dimension reduction method inferior to the stochastic subgradient method. In addition, the computational complexity of the dimension reduction method has little dependence on $N$. This is because we choose the separation oracle to be $(N\epsilon/4, \delta/4)$-$\mathcal{SO}$ and the computational complexity of each evaluation is independent of $N$.

## 7   Conclusions

In this work, we provide the first polynomial-time algorithms for the stochastic $L^\natural$-convex minimization problem, which serves as an important extension of the stochastic submodular minimization problem. Comparing to existing stochastic submodular minimization algorithms, our algorithms either achieve a lower computational complexity or do not require extra information about the objective function. We also derive a lower bound on the computational complexity to achieve the PAC guarantee. Numerical results are implemented to verify our theoretical findings.

## Acknowledgments and Disclosure of Funding

Haixiang Zhang and Javad Lavaei were supported by grants from AFOSR, ARO, ONR, NSF and C3.ai Digital Transformation Institute.

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
