# A Literature Review

In contrast to continuous optimization, where problems with the convex structure are studied in depth, most works on discrete stochastic optimization (Futschik & Pflug, 1995; Gutjahr & Pflug, 1996; Futschik & Pflug, 1997; Kleywegt et al., 2002; Semelhago et al., 2020) do not consider the convex structure. The main obstacle to the development of discrete convex optimization lies in the lack of a suitable definition of the discrete convex structure. A natural definition of the discrete convex functions would be functions that are extensible to continuous convex functions. However, for that class of functions, the local optimality does not imply the global optimality and therefore it is not suitable for the purpose of optimization. Later, Favati (1990) proposed a stronger condition, named the integral convexity, which ensures that the local optimality is equivalent to the global optimality. On the other hand, similar properties of continuous convex functions have been proved for submodular functions. Lovász (1983) showed the equivalence between the submodularity of a function and the convexity of its Lovász extension, and later Fujishige (1984) established the Fenchel-type min-max duality theorem for submodular functions. Moreover, the Lovász extension along with the subgradient convinced in Fujishige (2005) provide a good framework of applying gradient-based method to the submodular function minimization (SFM) problem. In the case when the objective function is an integer-valued submodular function, the best strongly and weakly polynomial algorithms proposed in Lee et al. (2015) respectively return the optimal solution with $O(d^3 \log^2(d))$ and $O(d^2 \log(Md))$ function value evaluations, where $d$ is the dimension and $M$ is the maximal absolute objective function value. The state-of-the-art approximate algorithm designed in (Axelrod et al., 2020) returns an $\epsilon$-approximate solution with $O(d \log(d)/\epsilon^2)$ function value evaluations. Our problem can be viewed as stochastic versions of the problems in Chakrabarty et al. (2017); Axelrod et al. (2020), which also naturally cause the difference in algorithm design and analysis. The efficiency of the algorithms in Chakrabarty et al. (2017); Axelrod et al. (2020) comes from on sampling a stochastic subgradient with $O(1)$ variance within $O(1)$ evaluations, which is based on the preprocessing phase (Lemma 3.4 in Axelrod et al. (2020)). However, the sampling phase requires an importance sampling based on the exact function values difference that is computed in the preprocessing phase. It is challenging to extend the importance sampling to the case when we only have access to noisy function values. The major techniques in our gradient-based algorithm is the truncation step, which is proved to reduce the sample complexity by $O(d)$. In Bach (2019); Axelrod et al. (2020), the authors extend the domain of submodular function to $\{1, 2, \ldots, N\} \times \cdots \times \{1, 2, \ldots, N\}$ and show that the generalized submodular function is equivalent to a submodular function on a sub-lattice of $\mathbb{Z}^{(N-1)d}$.

Instead of the deterministic SFM problem, the stochastic SFM problem is far from being well-understood. Recently, Ito (2019) provided upper bounds on the computational complexity for expectation error bounds. In addition, the authors of Ito (2019) proved a lower bound of computational complexity for achieving the given precision and showed that the stochastic SFM problem is essentially more difficult than the deterministic SFM problem. The first polynomial-time approximate algorithm to achieve a high-probability error bound is given in Blais et al. (2019). In this work, we also focus on finding solutions with a small error with high probability. We propose algorithms that either have a smaller computational complexity compared to the algorithms in Blais et al. (2019) or do not require information about the objective function. An alternative problem, the stochastic submodular maximization problem, has been studied in depth under the assumption that the objective function is monotone; see Hassani et al. (2017); Karimi et al. (2017); Hassidim & Singer (2017); Mokhtari et al. (2018); Sekar et al. (2021). We note that the approximate solutions to the maximization problem often include a multiplicative error instead of an additive error for the SFM problem. We refer the readers to Ito (2019) for a detailed discussion on related works for the stochastic SFM problem.

In Murota (2003), a generalization of submodular functions, called the $L^\natural$-convex functions, are defined through the translation submodularity or the discrete midpoint convexity. The $L^\natural$-convex functions are equivalent to functions that are both submodular and integrally convex on integer lattice. In addition, the $L^\natural$-convex function has a convex extension that shares similar properties as the Lovász extension and therefore gradient-based methods are also applicable for the $L^\natural$-convex function minimization problem. We refer the readers to Murota (2003) for more discussion on the discrete convex functions.

Since we are optimizing the Lovász extension of the submodular function without accessing its derivatives, we can view submodular minimization algorithms as zeroth-order convex optimization

algorithms. The idea of approximating gradient using finite difference appeared as early as Nemirovsky & Yudin (1983) and was elaborated in Nesterov & Spokoiny (2017) to derive bounds of zeroth-order methods on smooth and non-smooth convex problems. Later, Duchi et al. (2015) utilized this idea to design the optimal zeroth-order mirror descent method for stochastic convex problems. Furthermore, the RSGF method in Ghadimi & Lan (2013) also incorporates the Gaussian smoothing technique and gives the first bound on stochastic nonconvex smooth problems. In the follow-up work by Balasubramanian & Ghadimi (2018), the zeroth-order conditional gradient method is designed to handle the constrained case. In the same literature, a linear-time zeroth-order estimator of the Hessian matrix was proposed and was used to construct a saddle-point avoiding method. We refer the readers to Homem-de Mello & Bayraksan (2014); Larson et al. (2019) for a review of recent developments of zeroth-order algorithms.

The optimization problems via stochastic objective value evaluations have also been widely studied in the simulation literature. The problem is often called the ranking-and-selection (R&S); see Hong et al. (2020) for a recent review. There have been two approaches to categorize the R&S literature. One approach is differentiating the frequentist view and the Bayesian view; see Kim & Nelson (2006); Chick (2006). The other approach differentiates the fixed-confidence procedures and the fixed-budget procedures; see Frazier et al. (2009); Hunter & Nelson (2017); Hong et al. (2020). In particular, the probability of correct selection (PCS) of the best decision has been a widely used guarantee for both types of procedures. In general, the R&S procedures under fixed-precision and fixed-budget were often classified into the frequentist and Bayesian procedures in the literature. However, there are some exceptions Frazier (2014). A large number of R&S procedures based on the PCS guarantee adopt the indifference zone (IZ) formulation, called PCS-IZ. The IZ parameter is typically assumed to be known, while Fan et al. (2016), as a notable exception, provides selection guarantees without the knowledge of the indifference-zone parameter. In practice, for some problem settings, this IZ parameter may be unknown a priori. This naturally gives rise to a notion of probability of good selection (PGS), which is the optimality guarantee considered in this work. Eckman & Henderson (2018a,b) have thoroughly discussed settings when the use of PGS is preferable compared to the use of PCS-IZ. Generally in the R&S problems, there is no structural information such as convexity that is considered.

The optimization problems considered in this work also belong to the class of discrete optimization via simulation problems, the discussions of which can be found in Fu (2002); Nelson (2010); Sun et al. (2014); Hong et al. (2015); Chen et al. (2018). Discrete optimization via simulation problems naturally arise in many operations research and management science applications, including queueing networks, supply chain networks, sharing economy operations, financial markets, etc.; see Shaked & Shanthikumar (1988); Wolff & Wang (2002); Altman et al. (2003); Singhvi et al. (2015); Jian et al. (2016); Freund et al. (2017) for example. Hu et al. (2007, 2008) have discussed model reference adaptive search algorithms in order to ensure global convergence. Hong & Nelson (2006); Hong et al. (2010); Xu et al. (2010) propose and study algorithms based on the convergent optimization via most-promising-area stochastic search (COMPASS) that can be used to solve general simulation optimization problems with discrete decision variables. The proposed algorithms are computationally efficient and are proved to convergence with probability one to optimal points. Lim (2012) studies simulation optimization problems over multidimensional discrete sets where the objective function adopts multimodularity. They propose algorithms that converge almost surely to the global optimal. Wang et al. (2013) discusses stochastic optimization problems with integer-ordered decision variables. Park & Kim (2015) and Park et al. (2014) develop and examine the Penalty Function with Memory (PFM) method for discrete optimization via simulation with stochastic constraints. Sun et al. (2014) discusses an exploration-exploitation balancing approach using Gaussian process based search. Eckman et al. (2020) discusses a statistically guaranteed screening to rule out decisions based on initial simulation experiments utilizing the convex structure. Sekar et al. (2021) proposes an adaptive line search method that guarantees the convergence to local minima.

# B  Preliminaries on $L^\natural$-convex functions

In this section, we provide a detailed discussion on the properties of $L^\natural$-convex functions. The following property shows that $L^\natural$-convex functions can be viewed as a generalization of submodular functions.

**Lemma 1** (Murota (2003)). *Suppose that the function $f(x) : \mathcal{X} \mapsto \mathbb{R}$ is $L^\natural$-convex. Then, the following translation submodularity holds:*

$$f(x) + f(y) \geq f((x - \alpha\mathbf{1}) \vee y) + f(x \wedge (y + \alpha\mathbf{1})),$$
$$\forall x, y \in \mathcal{X}, \; \alpha \in \mathbb{N} \quad \text{s.t. } (x - \alpha\mathbf{1}) \vee y, \; x \wedge (y + \alpha\mathbf{1}) \in \mathcal{X}.$$

By the translation submodularity, the $L^\natural$-convex function restricted to a cube $x + \{0, 1\}^d \subset \mathcal{X}$ is a submodular function. Therefore, the Lovász extension (Lovász, 1983) can be constructed as the convex piecewise linear extension inside each cube. In addition, $L^\natural$-convex functions are integrally convex functions (Murota, 2003). Hence, we can obtain a continuous convex function on $[1, N]^d$ by piecing together the Lovász extension in each cube. More importantly, we can calculate a subgradient of the convex extension with $O(d)$ function value evaluations. Hence, $L^\natural$-convex functions provide a good framework for extending the continuous convex optimization theory to the discrete case. In the remainder of this subsection, we specify this intuition of $L^\natural$-convex functions in a rigorous way. We first define the Lovász extension of submodular functions and give an explicit subgradient of the Lovász extension at each point.

**Definition 2.** Suppose that $f(x) : \{0, 1\}^d \mapsto \mathbb{R}$ is a submodular function. For any $x \in [0, 1]^d$, we say that a permutation $\alpha_x : [d] \mapsto [d]$ is a **consistent permutation** of $x$, if

$$x_{\alpha_x(1)} \geq x_{\alpha_x(2)} \geq \cdots \geq x_{\alpha_x(d)}.$$

For each $i \in \{0, 1, \ldots, d\}$, the $i$-**th neighbouring points** of $x$ is defined as

$$S^{x,i} := \sum_{j=1}^{i} e_{\alpha_x(j)} \in \mathcal{X},$$

where vector $e_k$ is the $k$-th unit vector of $\mathbb{R}^d$. We define the **Lovász extension** $\tilde{f}(x) : [0, 1]^d \mapsto \mathbb{R}$ as

$$\tilde{f}(x) := f\left(S^{x,0}\right) + \sum_{i=1}^{d} \left[f\left(S^{x,i}\right) - f\left(S^{x,i-1}\right)\right] x_{\alpha_x(i)}. \tag{4}$$

We note that the value of the Lovász extension does not rely on the choice of the consistent permutation. We list several well-known properties of the Lovász extension and refer their proofs to Lovász (1983); Fujishige (2005).

**Lemma 2.** *The following properties hold for $\tilde{f}(x)$:*

  *(i) For any $x \in \mathcal{X}$, it holds that $\tilde{f}(x) = f(x)$.*

  *(ii) The minimizers of $\tilde{f}(x)$ satisfy $\arg\min_{x \in [0,1]^d} \tilde{f}(x) = \arg\min_{x \in \mathcal{X}} f(x)$.*

  *(iii) The function $\tilde{f}(x)$ is a convex function on $[0, 1]^d$.*

  *(iv) A subgradient $g \in \partial \tilde{f}(x)$ is given by*

$$g_{\alpha_x(i)} := f\left(S^{x,i}\right) - f\left(S^{x,i-1}\right), \quad \forall i \in [d]. \tag{5}$$

Using the expression (5), we construct a subgradient estimator at point $x$ as

$$\hat{g}_{\alpha_x(i)} := F\left(S^{x,i}, \xi_i^1\right) - F\left(S^{x,i-1}, \xi_{i-1}^2\right), \quad \forall i \in [d], \tag{6}$$

where $\xi_i^j$ are mutually independent for $i \in [d]$ and $j \in [2]$. By definition, we know the components of $\hat{g}$ are mutually independent and the computational complexity of each $\hat{g}$ is $2d$. Using the subgradient defined in (5), we have

$$\mathbb{E}\left[\hat{g}_{\alpha_x(i)}\right] = \mathbb{E}\left[F\left(S^{x,i}, \xi_i\right) - F\left(S^{x,i-1}, \xi_{i-1}\right)\right] = f\left(S^{x,i}\right) - f\left(S^{x,i-1}\right) = g_{\alpha_x(i)}, \quad \forall i \in [d],$$

which means that $\hat{g}$ is an unbiased estimator of $g$.

Then, we show that the Lovász extension in the neighborhood of each point can be pieced together to form a convex function on $\text{conv}(\mathcal{X}) = [1, N]^d$. We define the local neighborhood of each point $y \in [1, N - 1]^d$ as the cube

$$\mathcal{C}_y := y + [0, 1]^d,$$

where the Minkowski sum of a point $y \in \mathbb{R}^d$ and a set $\mathcal{C} \subset \mathbb{R}^d$ is defined as

$$y + \mathcal{C} := \{y + x \mid x \in \mathcal{C}\}.$$

We denote the objective function $f(x)$ restricted to $\mathcal{C}_y \cap \mathcal{X}$ as $f_y(x)$, which is submodular by the translation submodularity of $f(x)$. For point $x \in \mathcal{C}_y$, we denote $\alpha_x$ as a consistent permutation of $x - y$ in $\{0, 1\}^d$ and, for each $i \in \{0, 1, \ldots, d\}$, the corresponding $i$-th neighboring point of $x$ is defined as

$$S^{x,i} := y + \sum_{j=1}^{i} e_{\alpha_x(j)}.$$

Then, the Lovász extension of $f_y(x)$ in $\mathcal{C}_y$ can be calculated as

$$\tilde{f}_y(x) := f\left(S^{x,0}\right) + \sum_{i=1}^{d} \left[f\left(S^{x,i}\right) - f\left(S^{x,i-1}\right)\right] x_{\alpha_x(i)}.$$

Now, we piece together the Lovász extension in each cube by defining

$$\tilde{f}(x) := \tilde{f}_y(x) \quad \forall x \in [1, N]^d, \ y \in [N-1]^d, \quad \text{s.t. } x \in \mathcal{C}_y. \tag{7}$$

Using the results in Murota (2003), we can prove that $\tilde{f}(x)$ is well-defined and is a convex function.

**Theorem 8.** *The function $\tilde{f}(x)$ in (7) is well-defined and is convex on $\mathcal{X}$.*

Utilizing properties (i) and (ii) of Lemma 2, we know that problem (1) is equivalent to the relaxed problem

$$f^* := \min_{x \in [1,N]^d} \tilde{f}(x).$$

Moreover, the subgradient (5) and stochastic subgradient (6) are also valid for the convex extension $\tilde{f}(x)$. Similarly, (stochastic) subgradients can be computed in the neighboring cube of each point with $2d$ evaluations to $F(\cdot, \cdot)$ and it does not matter which cube is chosen for points belonging to multiple cubes.

### B.1 Illustrations of the Lovász extension

In this subsection, we show the Lovász extension of a two-dimensional function on $[3]^2 = \{1, 2, 3\}^2$. We consider the quadratic function

$$f(x) := x^T \begin{bmatrix} 0.101 & -0.068 \\ -0.068 & 0.146 \end{bmatrix} x, \quad \forall x \in \mathbb{R}^2.$$

By the results in Murota (2003, Section 7.3), we know the function $f(\cdot)$ is an $L^\natural$-convex function. We compare the landscapes of the original objective and the Lovász extension in Figure 1. We can see that the Lovász extension is a piecewise linear and convex function, which is consistent with the results in Murota (2003).

## C  Proofs in Section 3

### C.1  Proof of Theorem 2

*Proof of Theorem 2.* To gain some intuition, we first consider the deterministic case. Suppose that we already have an $\epsilon$-optimal solution to problem (2), i.e., a point $\bar{x}$ in $[1, N]^d$ such that $\tilde{f}(\bar{x}) \leq f^* + \epsilon$. Then, we rewrite the Lovász extension in (4) as

$$\tilde{f}(\bar{x}) = \left[1 - \bar{x}_{\alpha_{\bar{x}}(1)}\right] f\left(S^{\bar{x},0}\right) + \sum_{i=1}^{d-1} \left[\bar{x}_{\alpha_{\bar{x}}(i)} - \bar{x}_{\alpha_{\bar{x}}(i+1)}\right] f\left(S^{\bar{x},i}\right) + \bar{x}_{\alpha_{\bar{x}}(d)} f\left(S^{\bar{x},d}\right), \tag{8}$$

which is a convex combination of $f\left(S^{\bar{x},0}\right), \ldots, f\left(S^{\bar{x},d}\right)$. Hence, there exists an $\epsilon$-optimal solution among the neighboring points of $\bar{x}$. This means that we can solve a sub-problem with $d + 1$ points to get the $\epsilon$-optimal solution among neighboring points.

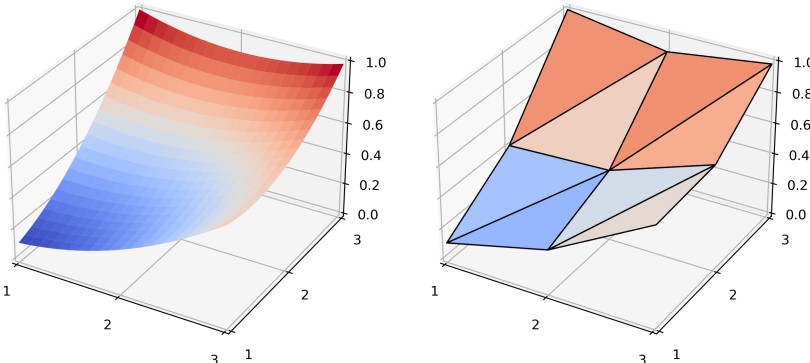

Figure 1: The landscapes of the objective function and its Lovász extension. **Left:** The original objective function. **Right:** The Lovász extension of the objective function.

The proof for the stochastic case is similar. We denote the optimal value of $f(x)$ as $f^*$. Since point $\bar{x}$ satisfies the $(\epsilon/2, \delta/2)$-PAC guarantee, we have

$$\tilde{f}(\bar{x}) - f^* \leq \epsilon/2$$

holds with probability at least $1 - \delta/2$. We assume this event happens in the following of this proof. Let $S^0, S^1, \ldots, S^d$ be the neighboring points of $\bar{x}$. Using the expression of the Lovász extension in (8), we know there exists an $\epsilon/2$-optimal solution among $S^0, S^1, \ldots, S^d$. We denote the $\epsilon/2$-optimal solution and the solution returned by Algorithm 1 as $S^*$ and $\hat{S}$, respectively. By the definition of confidence intervals, we know

$$\left|\hat{F}_n(S_i) - f(S_i)\right| \leq \epsilon/4 \quad \forall i \in \{0, \ldots, d\}, \quad \left|\hat{F}_n(\hat{S}) - f(\hat{S})\right| \leq \epsilon/4$$

holds uniformly with probability at least $1 - \delta/2$. Under this event, we know

$$f(\hat{S}) - f^* \leq \hat{F}_n(\hat{S}) - f^* + \epsilon/4 \leq \hat{F}_n(S^*) - f^* + \epsilon/4 \leq f(S^*) - f^* + \epsilon/2 \leq \epsilon,$$

which implies that $x^* \in \mathcal{X}$ is an $\epsilon$-optimal solution and the probability is at least $1 - \delta/2 - \delta/2 = 1 - \delta$. Hence, we know $x^*$ is an $(\epsilon, \delta)$-PAC solution to problem (1).

Now, we estimate the computational complexity of Algorithm 1. By the Hoeffding bound, simulating

$$\frac{32}{\epsilon^2} \log\left(\frac{8d}{\delta}\right)$$

times on each neighboring point is enough to achieve $1 - \delta/(4d)$ confidence half-width $\epsilon/4$. Hence, the computational complexity of Algorithm 1 is at most

$$\frac{32(d+1)}{\epsilon^2} \log\left(\frac{8d}{\delta}\right) = O\left[\frac{d}{\epsilon^2} \log\left(\frac{d}{\delta}\right)\right] = \tilde{O}\left[\frac{d}{\epsilon^2} \log\left(\frac{1}{\delta}\right)\right].$$

$\square$

### C.2 Proof of Theorem 3

The analysis of Algorithm 2 fits into the classical convex optimization framework. The following Azuma's inequality for martingales with sub-Gaussian tails plays as a major role for deriving high-probability bounds, i.e., the number of required samples to ensure the algorithms succeed with high probability.

**Lemma 3** (Azuma's inequality for sub-Gaussian tails (Shamir, 2011)). *Let $X_0, \ldots, X_{T-1}$ be a martingale difference sequence. Suppose there exist constants $b_1 \geq 1, b_2 > 0$ such that, for any $t \in \{0, \ldots, T-1\}$,*

$$\mathbb{P}(|X_t| \geq a \mid X_1, \ldots, X_{t-1}) \leq 2b_1 \exp(-b_2 a^2), \quad \forall a \geq 0. \tag{9}$$

*Then for any $\delta > 0$, it holds with probability at least $1 - \delta$ that*

$$\frac{1}{T} \sum_{t=0}^{T-1} X_t \leq \sqrt{\frac{28 b_1}{b_2 T} \log(\frac{1}{\delta})}.$$

Since the stochastic subgradient $\hat{g}^t$ is truncated, the stochastic subgradient used for updating, namely $\tilde{g}^t$, is not unbiased. We define the bias at each step as

$$b_t := \mathbb{E}\left[ \tilde{g}^t \mid x^0, x^1, \ldots, x^t \right] - g^t, \quad \forall t \in \{0, 1, \ldots, T-1\}.$$

First, we bound the $\ell_1$-norm of the bias.

**Lemma 4.** *Suppose Assumptions 1-4 hold. If we have*

$$M \geq 2\sigma \cdot \sqrt{\log(\frac{8\sigma dT}{\epsilon})} = \Theta\left[ \sqrt{\log(\frac{dT}{\epsilon})} \right], \quad T \geq \frac{2\epsilon}{\sigma},$$

*then it holds*

$$\|b^t\|_1 \leq \frac{\epsilon}{2T}, \quad \forall t \in \{0, 1, \ldots, T-1\}.$$

*Proof.* Let $\alpha_t$ be a consistent permutation of $x^t$ and $S^{t,i}$ be the corresponding $i$-th neighboring points in the neighbourhood of $x^t$. We only need to prove

$$\left| b^t_{\alpha_t(i)} \right| \leq \frac{\epsilon}{2dT}, \quad \forall i \in [d].$$

We define two random variables

$$Y_1 := F\left( S^{t,i}, \xi^1_i \right) - f\left( S^{t,i} \right), \quad Y_2 := F\left( S^{t,i-1}, \xi^2_{i-1} \right) - f\left( S^{t,i-1} \right).$$

By Assumption 1, both $Y_1$ and $Y_2$ are independent and sub-Gaussian with parameter $\sigma^2$. Hence, we know

$$b^t_{\alpha_t(i)} = \mathbb{E}\left[ \tilde{g}^t_{\alpha_t(i)} - g^t_{\alpha_t(i)} \right] = \mathbb{E}\left[ (Y_1 + Y_2) \cdot \mathbf{1}_{-M \leq Y_1 + Y_2 \leq M} \right] + \mathbb{E}\left[ M \cdot \mathbf{1}_{Y_1 + Y_2 > M} \right] + \mathbb{E}\left[ -M \cdot \mathbf{1}_{Y_1 + Y_2 < -M} \right]$$

$$= \mathbb{E}\left[ (M - Y_1 - Y_2) \cdot \mathbf{1}_{Y_1 + Y_2 > M} \right] + \mathbb{E}\left[ -(M + Y_1 + Y_2) \cdot \mathbf{1}_{Y_1 + Y_2 < -M} \right],$$

where the second step is from $\mathbb{E}[Y_1] = \mathbb{E}[Y_2] = 0$. Taking the absolute value on both sides, we get

$$\left| b^t_{\alpha_t(i)} \right| \leq \mathbb{E}\left[ (Y_1 + Y_2 - M) \cdot \mathbf{1}_{Y_1 + Y_2 > M} \right] + \mathbb{E}\left[ -(M + Y_1 + Y_2) \cdot \mathbf{1}_{Y_1 + Y_2 < -M} \right] \qquad (10)$$

$$= \mathbb{E}\left[ (Y - M) \cdot \mathbf{1}_{Y > M} \right] + \mathbb{E}\left[ -(Y + M) \cdot \mathbf{1}_{Y < -M} \right],$$

where we define the random variable $Y := Y_1 + Y_2$. Since $Y_1, Y_2$ are independent, random variable $Y$ is sub-Gaussian with parameter $2\sigma^2$. Let $F(y) := \mathbb{P}[Y \leq y]$ be the distribution function of $Y$. Then, we have

$$\mathbb{E}\left[ (Y - M) \cdot \mathbf{1}_{Y > M} \right] = \int_M^\infty (y - M)\, dF(y) = \int_M^\infty (1 - F(y))\, dy. \qquad (11)$$

By the Hoeffding bound, we know

$$1 - F(y) = \mathbb{P}[Y > y] \leq \exp\left( -y^2/4\sigma^2 \right), \quad \forall y \geq 0.$$

Using the upper bound for $Q$-function in Borjesson & Sundberg (1979), it holds that

$$\int_M^\infty 1 - F(y)\, dy \leq \int_M^\infty \exp\left( -y^2/4\sigma^2 \right)\, dy \leq \frac{2\sigma^2}{M} \exp\left( -\frac{M^2}{4\sigma^2} \right).$$

By the choice of $M$, we know

$$M \geq 2\sigma\sqrt{\log(8d)} \geq 2\sigma \quad \text{and} \quad \sigma \exp(-M^2/4\sigma^2) \leq \frac{\epsilon}{4dT}.$$

which implies that

$$\int_M^\infty 1 - F(y)\, dy \leq \frac{2\sigma^2}{M} \exp(-M^2/4\sigma^2) \leq \frac{\epsilon}{4dT}.$$

Substituting the above inequality into (11), we have

$$\mathbb{E}\left[(Y - M) \cdot \mathbf{1}_{Y > M}\right] \leq \frac{\epsilon}{4dT}.$$

Considering $-Y$ in the same way, we can prove

$$\mathbb{E}\left[-(Y + M) \cdot \mathbf{1}_{Y < -M}\right] \leq \frac{\epsilon}{4dT}.$$

Substituting the last two estimates into inequality (10), we know

$$\left| b_{\alpha_t(i)}^t \right| \leq \frac{\epsilon}{2dT}.$$

$\square$

Next, we show that $\langle g^t + b^t - \tilde{g}^t, x^t - x^* \rangle$ forms a martingale sequence and use Azuma's inequality to bound the deviation, where $x^*$ is a minimizer of $f(x)$.

**Lemma 5.** *Suppose Assumptions 1-4 hold and let $x^*$ be a minimizer of $f(x)$. The sequence*

$$X_t := \left\langle g^t + b^t - \tilde{g}^t, x^t - x^* \right\rangle \quad t = 0, 1, \ldots, T - 1$$

*forms a martingale difference sequence. Furthermore, if we have*

$$M = \max\left\{ L, 2\sigma \cdot \sqrt{\log(\frac{4\sigma dNT}{\epsilon})} \right\} = \tilde{\Theta}\left[ \sqrt{\log(\frac{dNT}{\epsilon})} \right], \quad T \geq \frac{2\epsilon}{\sigma},$$

*then it holds*

$$\frac{1}{T} \sum_{t=0}^{T-1} X_t \leq \sqrt{\frac{224 dN^2 \sigma^2}{T} \log(\frac{1}{\delta})}$$

*with probability at least $1 - \delta$.*

*Proof.* Let $\mathcal{F}_t$ be the filtration generated by $x_0, x_1, \ldots, x_t$. By the definition of $b^t$, we know

$$\mathbb{E}\left[g^t + b^t - \tilde{g}^t \mid \mathcal{F}_t\right] = 0,$$

which implies that

$$\mathbb{E}\left[X_t \mid \mathcal{F}_t\right] = \left\langle \mathbb{E}\left[g^t + b^t - \tilde{g}^t \mid \mathcal{F}_t\right], x^t - x^* \right\rangle = 0.$$

Hence, the sequence $\{X_t\}$ is a martingale difference sequence. Next, we estimate the probability $\mathbb{P}[|X_t| \geq a \mid \mathcal{F}_t]$. We have the bound

$$|X_t| = \left| \left\langle g^t + b^t - \tilde{g}^t, x^t - x^* \right\rangle \right| \leq \left\| g^t + b^t - \tilde{g}^t \right\|_1 \left\| x^t - x^* \right\|_\infty$$
$$\leq N \left\| g^t + b^t - \tilde{g}^t \right\|_1 \leq N \left\| g^t - \tilde{g}^t \right\|_1 + N \left\| b^t \right\|_1.$$

Since $M$ satisfies the condition in Lemma 4, we know $\|b^t\|_1 \leq \epsilon/2T$. Recalling Assumption 4, we get $|g_i^t| \leq L$ for all $i \in [d]$. By the truncation rule and the assumption $M \geq L$, we have

$$\left| \tilde{g}_i^t - g_i^t \right| = \left| (\hat{g}_i^t \wedge M) \vee (-M) - g_i^t \right| \leq \left| \hat{g}_i^t - g_i^t \right|, \quad \forall i \in [d].$$

Hence, we get

$$|X_t| \leq \frac{N\epsilon}{2T} + N \left\| \hat{g}^t - g^t \right\|_1. \tag{12}$$

Define random variables $Y_i := |\hat{g}_i^t - g_i^t|$ for all $i \in [d]$. By Assumption 1, $Y_i$ is sub-Gaussian with parameter $\sigma^2$. Hence, we have

$$Y := \left\| \hat{g}^t - g^t \right\|_1 = \sum_{i=1}^d Y_i$$

is sub-Gaussian with parameter $d\sigma^2$. First, we consider the case when $a/N \geq \epsilon/T$. Using inequality (12), it follows that

$$\mathbb{P}\left[|X_t| \geq a \mid \mathcal{F}_\sigma\right] \leq \mathbb{P}\left[\frac{\epsilon}{2T} + Y \geq \frac{a}{N}\right] \leq \mathbb{P}\left[Y \geq \frac{a}{N} - \frac{\epsilon}{2T}\right]$$

$$\leq \mathbb{P}\left[Y \geq \frac{a}{2N}\right] \leq 2\exp\left(-\frac{a^2}{8dN^2\sigma^2}\right), \tag{13}$$

where the last inequality is from Hoeffding bound. In this case, we know condition (9) holds with

$$b_1 = 1, \quad b_2 = \frac{1}{8dN^2\sigma^2}.$$

Now, we consider the case when $a/N < \epsilon/T$. In this case, by the assumption that $T \geq 2\epsilon/\sigma$, we have

$$2b_1 \exp\left(-b_2 a^2\right) > 2\exp\left(-\frac{1}{8dN^2\sigma^2}\cdot\frac{\epsilon^2}{T^2}\right) \geq 2\exp\left(-\frac{1}{32d}\right) \geq 2\exp\left(-\frac{1}{32}\right) > 1.$$

Hence, it holds

$$\mathbb{P}\left[|X_t| \geq a \mid \mathcal{F}_\sigma\right] \leq 1 < 2b_1 \exp\left(-b_2 a^2\right).$$

Combining with inequality (13), we know condition (9) holds with $b$ and $c$ defined above. Using Lemma 3, we know

$$\frac{1}{T}\sum_{t=0}^{T-1} X_t \leq \sqrt{\frac{224dN^2\sigma^2}{T}\log(\frac{1}{\delta})}$$

holds with probability at least $1 - \delta$. $\qquad\square$

Then, we prove a lemma similar to the Lemma in Zinkevich (2003).

**Lemma 6.** *Suppose Assumptions 1-4 hold and let $x^*$ be a minimizer of $f(x)$. If we choose*

$$\eta = \frac{N}{M\sqrt{T}},$$

*then we have*

$$\frac{1}{T}\sum_{t=0}^{T-1} \langle \tilde{g}^t, x^t - x^* \rangle \leq \frac{dNM}{\sqrt{T}}.$$

*Proof.* We define $\tilde{x}^{t+1} := x^t - \eta\tilde{g}^t$ as the next point before the projection onto $[1, N]^d$. Recalling the non-expansion property of orthogonal projection, we get

$$\|x^{t+1} - x^*\|_2^2 = \|\mathcal{P}\left(\tilde{x}^{t+1} - x^*\right)\|_2^2 \leq \|\tilde{x}^{t+1} - x^*\|_2^2 = \|x^t - x^* - \eta\tilde{g}^t\|_2^2$$
$$= \|x^t - x^*\|_2^2 + \eta^2\|\tilde{g}^t\|_2^2 - 2\eta\langle \tilde{g}^t, x^t - x^* \rangle,$$

and equivalently,

$$\langle \tilde{g}^t, x^t - x^* \rangle = \frac{1}{2\eta}\left[\|x^t - x^*\|_2^2 - \|x^{t+1} - x^*\|_2^2\right] + \frac{\eta}{2}\cdot\|\tilde{g}^t\|_2^2.$$

Summing over $t = 0, 1, \ldots, T-1$, we have

$$\sum_{t=0}^{T-1}\langle \tilde{g}^t, x^t - x^* \rangle = \frac{\|x^0 - x^*\|_2^2 - \|x^T - x^*\|_2^2}{2\eta} + \frac{\eta}{2}\sum_{t=0}^{T-1}\|\tilde{g}^t\|_2^2$$
$$\leq \frac{d\|x^0 - x\|_\infty^2}{2\eta} + \frac{\eta}{2}\sum_{t=0}^{T-1}\|\tilde{g}^t\|_2^2 \leq \frac{dN^2}{2\eta} + \frac{\eta}{2}\sum_{t=0}^{T-1}\|\tilde{g}^t\|_2^2.$$

By the definition of truncation, it follows that $\|\tilde{g}^t\|_2^2 \leq dM^2$. Choosing

$$\eta := \frac{N}{M\sqrt{T}},$$

it follows that

$$\sum_{t=0}^{T-1}\langle \tilde{g}^t, x^t - x^* \rangle \leq \frac{dN^2}{2\eta} + \frac{\eta}{2}\sum_{t=0}^{T-1}\|\tilde{g}^t\|_2^2 \leq \frac{dN^2}{2\eta} + \frac{\eta TdM^2}{2} = dNM\sqrt{T}.$$

$\qquad\square$

Finally, using Lemmas 4, 5 and 6, we can finish the proof of Theorem 3.

*Proof of Theorem 3.* Denote $f^*$ as the optimal value of $\tilde{f}(x)$. Using the convexity of $\tilde{f}(x)$, we know

$$\tilde{f}(\bar{x}) - f^* \leq \frac{1}{T} \sum_{t=0}^{T-1} \left[ \tilde{f}(x^t) - f^* \right] \leq \frac{1}{T} \sum_{t=0}^{T-1} \langle g^t, x^t - x^* \rangle \tag{14}$$

$$= \frac{1}{T} \sum_{t=0}^{T-1} \left[ \langle g^t + b^t - \tilde{g}^t, x^t - x^* \rangle + \langle \tilde{g}^t, x^t - x^* \rangle - \langle b^t, x^t - x^* \rangle \right].$$

We choose

$$T := \frac{3584 d N^2 \sigma^2}{\epsilon^2} \log(\frac{2}{\delta}) = \Theta \left[ \frac{d N^2}{\epsilon^2} \log(\frac{1}{\delta}) \right].$$

Recalling Assumption 1, we know $\delta$ is small enough and therefore we have the following estimates:

$$L^2 \leq M^2 = \tilde{\Theta} \left[ \log(\frac{dT}{\epsilon}) \right] = \tilde{O} \left[ \log(\frac{d^2 N^2}{\epsilon^3}) + \log\log(\frac{1}{\delta}) \right] \leq \frac{\epsilon^2 T}{64 d^2}, \quad T \geq \max \left\{ \frac{2\epsilon}{\sigma}, 4 \right\}.$$

Hence, the conditions in Lemmas 4 and 5 are satisfied. By Lemma 4, we know

$$-\frac{1}{T} \sum_{t=0}^{T-1} \langle b^t, x^t - x^* \rangle \leq \frac{1}{T} \sum_{t=0}^{T-1} \|b^t\|_1 \|x^t - x^*\|_\infty \leq \frac{\epsilon}{2T} \leq \frac{\epsilon}{8}. \tag{15}$$

By Lemma 5, it holds

$$\frac{1}{T} \sum_{t=0}^{T-1} \langle g^t + b^t - \tilde{g}^t, x^t - x^* \rangle \leq \sqrt{\frac{224 d N^2 \sigma^2}{T} \log(\frac{2}{\delta})} \leq \frac{\epsilon}{4} \tag{16}$$

with probability at least $1 - \delta$, where the last inequality is from our choice of $T$. By Lemma 6, we know

$$\frac{1}{T} \sum_{t=0}^{T-1} \langle \tilde{g}^t, x^t - x^* \rangle \leq \frac{d N M}{\sqrt{T}} \leq \frac{\epsilon}{8}. \tag{17}$$

Substituting inequalities (15), (16) and (17) into inequality (14), we get

$$\tilde{f}(\bar{x}) - f^* \leq \frac{\epsilon}{2}$$

holds with probability at least $1 - \delta/2$. By the results of Theorem 2, we know Algorithm 2 returns an $(\epsilon, \delta)$-PAC solution.

Finally, we estimate the computational complexity of Algorithm 2. For each iteration, we need to generate a stochastic subgradient using (6) and the computational complexity is $2d$. Hence, the total computational complexity of all iterations is

$$2d \cdot T = \tilde{\Theta} \left[ \frac{d^2 N^2}{\epsilon^2} \log(\frac{1}{\delta}) \right].$$

By Theorem 2, the computational complexity of rounding process is at most

$$\tilde{O} \left[ \frac{d}{\epsilon^2} \log(\frac{1}{\delta}) \right].$$

Thus, we know the total computational complexity of Algorithm 2 is at most

$$\tilde{O} \left[ \frac{d^2 N^2}{\epsilon^2} \log(\frac{1}{\delta}) \right].$$

$\square$

### C.3 Details of examples where Assumption 5 holds

**Example 3.** We consider the case when the randomness of each choice of decision variables shares the same measure space, i.e., there exists a measure space $(Z, \mathcal{B}_Z)$ such that $\xi_x$ can be any element in the measure space for all $x \in \mathcal{X}$. Moreover, for any fixed $\xi \in \mathcal{B}$, the function $F(\cdot, \xi)$ is also $L^\natural$-convex (or submodular when $N = 2$) and has $\ell_\infty$-Lipschitz constant $\tilde{L}$. Then, we consider the subgradient estimator

$$\hat{g}_{\alpha_x(i)} := F\left(S^{x,i}, \xi\right) - F\left(S^{x,i-1}, \xi\right) \quad \forall i \in [d]. \tag{18}$$

The computational complexity of estimator (18) is $d + 1$. In addition, property (v) of Lemma 2 gives

$$\|\hat{g}\|_1 \leq 3\tilde{L}/2.$$

Therefore, in this situation, the Assumption 5 holds with $G = 3\tilde{L}/2$ and $\beta = d + 1$.

When the distribution at each choice of decision variables is bounded almost surely, we show that Assumption 5 also holds.

**Example 4.** We consider the case when the distribution at each point $x \in \mathcal{X}$ is bounded, namely, there exists a constant $C \geq 0$ such that

$$\mathbb{P}[|F(x, \xi_x)| \leq C] = 1, \quad \forall x \in \mathcal{X}.$$

We note that a bounded distribution is a special case of sub-Gaussian distributions. In this case, the $\ell_\infty$-Lipschitz constant is $2C$ and property (v) in Lemma 2 gives $\|g\|_1 \leq 3C$ for any subgradient $g$. We consider the subgradient estimator (6). At point $x$, if index $i$ is chosen, then we know that

$$\|\hat{g}\|_1 = d \cdot \left|F\left(S^{x,i}, \xi_i^1\right) - F\left(S^{x,i-1}, \xi_{i-1}^2\right)\right| \leq 2dC.$$

Hence, Assumption 5 holds with $G = 2dC$ and $\beta = 2$.

### C.4 Proof of Theorem 4

Since the stochastic gradient is bounded, we apply the Azuma's inequality for martingale difference sequences with bounded tails.

**Lemma 7** (Azuma's inequality with bounded tails)**.** *Let $X_0, \ldots, X_{T-1}$ be a martingale difference sequence. Suppose there exists a constant $b$ such that for any $t \in \{0, \ldots, T-1\}$,*

$$\mathbb{P}(|X_t| \leq b) = 1.$$

*Then for any $\delta > 0$, it holds with probability at least $1 - \delta$ that*

$$\frac{1}{T} \sum_{t=0}^{T-1} X_t \leq b \sqrt{\frac{2}{T} \log(\frac{1}{\delta})}. \tag{19}$$

The proof follows a similar way as Theorem 3. We first bound the noise term by Azuma's inequality.

**Lemma 8.** *Suppose Assumptions 1-5 hold and let $x^*$ be a minimizer of $f(x)$. Then, it holds*

$$\frac{1}{T} \sum_{t=0}^{T-1} \left\langle g^t - \hat{g}^t, x^t - x^* \right\rangle \leq N\left(\frac{3L}{2} + G\right) \sqrt{\frac{2}{T} \log(\frac{1}{\delta})}$$

*with probability at least $1 - \delta$.*

*Proof.* Same as the proof of Lemma 5, the fact that $\hat{g}^t$ is unbiased implies that

$$X_t := \left\langle g^t - \hat{g}^t, x^t - x^* \right\rangle \quad t = 0, 1, \ldots, T-1$$

is a martingale difference sequence. By Assumption 5 and property (v) in Lemma 2, we know

$$|X_t| = \left|\left\langle g^t - \hat{g}^t, x^t - x^* \right\rangle\right| \leq \left\|g^t - \hat{g}^t\right\|_1 \left\|x^t - x^*\right\|_\infty \leq N \left\|g^t - \hat{g}^t\right\|_1 \leq N(3L/2 + G),$$

which implies that the condition (19) holds with $b = N(3L/2 + G)$. Using Lemma 7, we get the conclusion of this lemma. $\qquad\square$

The following lemma bounds the error of the algorithm and is similar to Theorem 3.2.2 in Nesterov (2018).

**Lemma 9.** *Suppose Assumptions 1-5 hold and let $x^*$ be a minimizer of $f(x)$. If we choose*

$$\eta = \sqrt{\frac{dN^2}{TG^2}},$$

*then we have*

$$\frac{1}{T}\sum_{t=0}^{T-1}\left\langle \hat{g}^t, x^t - x^*\right\rangle \le \sqrt{\frac{dN^2G^2}{T}}.$$

*Proof.* We define $\tilde{x}^{t+1} := x^t - \eta\hat{g}^t$ as the next point before the projection onto $[1, N]^d$. Recalling the non-expansion property of orthogonal projection, we get

$$\|x^{t+1} - x^*\|_2^2 = \|\mathcal{P}\left(\tilde{x}^{t+1} - x^*\right)\|_2^2 \le \|\tilde{x}^{t+1} - x^*\|_2^2 = \|x^t - x^* - \eta\hat{g}^t\|_2^2$$
$$= \|x^t - x^*\|_2^2 + \eta^2\|\tilde{g}^t\|_2^2 - 2\eta\langle\hat{g}^t, x^t - x^*\rangle,$$

and equivalently,

$$\langle\hat{g}^t, x^t - x^*\rangle = \frac{1}{2\eta}\left[\|x^t - x^*\|_2^2 - \|x^{t+1} - x^*\|_2^2\right] + \frac{\eta}{2}\cdot\|\hat{g}^t\|_2^2.$$

Using Assumption 5, we know $\|\hat{g}^t\|_2^2 \le \|\hat{g}^t\|_1^2 \le G^2$ and therefore

$$\langle\hat{g}^t, x^t - x^*\rangle = \frac{1}{2\eta}\left[\|x^t - x^*\|_2^2 - \|x^{t+1} - x^*\|_2^2\right] + \frac{\eta G^2}{2}.$$

Summing over $t = 0, 1, \ldots, T - 1$, we have

$$\sum_{t=0}^{T-1}\langle\hat{g}^t, x^t - x^*\rangle = \frac{\|x^0 - x^*\|_2^2 - \|x^T - x^*\|_2^2}{2\eta} + T\cdot\frac{\eta G^2}{2}$$

$$\le \frac{d\|x^0 - x\|_\infty^2}{2\eta} + \frac{\eta TG^2}{2} \le \frac{dN^2}{2\eta} + \frac{\eta TG^2}{2}.$$

Choosing

$$\eta := \sqrt{\frac{dN^2}{TG^2}},$$

it follows that

$$\sum_{t=0}^{T-1}\langle\tilde{g}^t, x^t - x^*\rangle \le NG\sqrt{dT}.$$

$\square$

Now, we prove Theorem 4 using Lemmas 8 and 9.

*Proof of Theorem 4.* According to to the proof of Theorem 3, we have

$$\tilde{f}(\bar{x}) - f^* \le \frac{1}{T}\sum_{t=0}^{T-1}\left[\tilde{f}(x^t) - f^*\right] \le \frac{1}{T}\sum_{t=0}^{T-1}\langle g^t, x^t - x^*\rangle \qquad (20)$$

$$= \frac{1}{T}\sum_{t=0}^{T-1}\langle\hat{g}^t, x^t - x^*\rangle + \frac{1}{T}\sum_{t=0}^{T-1}\langle g^t - \hat{g}^t, x^t - x^*\rangle.$$

By Lemmas 8 and 9, it holds

$$\frac{1}{T}\sum_{t=0}^{T-1}\langle\hat{g}^t, x^t - x^*\rangle \le N\left(\frac{3L}{2} + G\right)\sqrt{\frac{2}{T}\log(\frac{2}{\delta})}, \quad \frac{1}{T}\sum_{t=0}^{T-1}\langle g^t - \hat{g}^t, x^t - x^*\rangle \le \sqrt{\frac{dN^2G^2}{T}}$$

with probability at least $1 - \delta/2$. Choosing

$$T = N^2 \left( \frac{3L}{2} + G \right)^2 \cdot \frac{32}{\epsilon^2} \log(\frac{2}{\delta}) = \Theta \left[ \frac{N^2(L+G)^2}{\epsilon^2} \log(\frac{1}{\delta}) \right],$$

we know

$$T \geq \frac{16 d N^2 G^2}{\epsilon^2}$$

when $\delta$ is small enough. Hence, we have

$$\frac{1}{T} \sum_{t=0}^{T-1} \left\langle \hat{g}^t, x^t - x^* \right\rangle \leq \frac{\epsilon}{4}, \quad \frac{1}{T} \sum_{t=0}^{T-1} \left\langle g^t - \hat{g}^t, x^t - x^* \right\rangle \leq \frac{\epsilon}{4}$$

holds with probability at least $1 - \delta/2$. Substituting into inequality (20), we have

$$\tilde{f}(\bar{x}) - f^* \leq \frac{\epsilon}{2}$$

holds with probability at least $1 - \delta/2$. By the results of Theorem 2, we know Algorithm 2 returns an $(\epsilon, \delta)$-PAC solution.

Finally, we estimate the computational complexity of Algorithm 2. For each iteration, the computational complexity is decided by the generation of a stochastic subgradient, which is at most $\beta$ by Assumption 5. Hence, the total computational complexity of all iterations is

$$O\left[\beta T\right] = \tilde{O} \left[ \frac{\beta N^2(L+G)^2}{\epsilon^2} \log(\frac{1}{\delta}) \right].$$

By Theorem 2, the computational complexity of rounding process is at most

$$\tilde{O} \left[ \frac{d}{\epsilon^2} \log(\frac{1}{\delta}) \right].$$

Thus, we know the total computational complexity of Algorithm 2 is at most

$$\tilde{O} \left[ \frac{\beta N^2(L+G)^2 + d}{\epsilon^2} \log(\frac{1}{\delta}) \right].$$

$\square$

### C.5 Algorithms for the known sub-optimality gap case

In this section, we consider the case when the global optimum is unique and the sub-optimality is known to be at least $c > 0$. In this case, it suffices to find a $(c, \delta)$-PAC solution. We first prove that the existence of sub-optimality is equivalent to the so-called weak sharp minima condition of the convex extension. In addition, we use the $\ell_\infty$ norm in place of the $\ell_2$ norm since the feasible set is a hypercube.

**Definition 3.** We say that a function $f(x) : \mathcal{X} \mapsto \mathbb{R}$ satisfies the **Weak Sharp Minimum (WSM)** condition, if the function $f(x)$ has a unique minimizer $x^*$ and there exists a constant $\kappa > 0$ such that

$$\|x - x^*\|_\infty \leq \kappa \left( f(x) - f^* \right), \quad \forall x \in \mathcal{X},$$

where $f^* := f(x^*)$.

The WSM condition was first defined in Burke & Ferris (1993), and is also called the polyhedral error bound condition in recent literature (Yang & Lin, 2018). In addition, the WSM condition is a special case of the global growth condition in Xu et al. (2016) with $\theta = 1$. The WSM condition can be used to leverage the distance between intermediate solutions and $(c, \delta)$-PAC solutions. Given a constant $c > 0$, we use $\mathcal{MC}_c$ to denote the set of $L^\natural$-convex models with sub-optimality gap at least $c$. The next theorem verifies that the WSM condition is equivalent to the existence of sub-optimality gap $c$.

**Theorem 9.** *Suppose that function $f(x) : \mathcal{X} \mapsto \mathbb{R}$ is an $L^\natural$-convex function and $\tilde{f}(x)$ is its Lovász extension on $[1, N]^d$. For any constant $c > 0$, function $f(x) \in \mathcal{MC}_c$ if and only if $\tilde{f}(x)$ satisfies the WSM condition with $\kappa = c^{-1}$.*

*Proof.* We first prove the sufficiency part and then consider the necessity part.

**Sufficiency.** Suppose there exists a constant $\kappa > 0$ such that the function $\tilde{f}(x)$ satisfies the WSM condition with $\kappa$. Considering any point $x \in \mathcal{X}\backslash\{x^*\}$, we know $\|x - x^*\|_\infty \geq 1$ and, by the WSM condition,

$$f(x) - f^* = \tilde{f}(x) - f^* \geq \kappa^{-1}\|x - x^*\|_\infty \geq \kappa^{-1}.$$

Thus, we know the sub-optimality gap for $f(x)$ is at least $\kappa^{-1}$ and $f(x) \in \mathcal{MC}_{\kappa^{-1}}$.

**Necessity.** Suppose there exists a constant $c > 0$ such that

$$f(x) - f^* \geq c \quad \forall x \in \mathcal{X}\backslash\{x^*\}.$$

We first consider point $x \in [1, N]^d$ such that $\|x - x^*\|_\infty \leq 1$. In this case, we know there exists a hypercube $\mathcal{C}_y$ containing both $x$ and $x^*$. By the definition of Lovász extension, we know that

$$\tilde{f}(x) = [1 - x_{\alpha_x(1)}]f\left(S^{x,0}\right) + \sum_{i=1}^{d-1} [x_{\alpha_x(i)} - x_{\alpha_x(i+1)}]f\left(S^{x,i}\right) + x_{\alpha_x(d)}f\left(S^{x,d}\right) = \sum_{i=0}^{d} \lambda_i f\left(S^{x,i}\right),$$

where we define

$$\lambda_i := x_{\alpha_x(i)} - x_{\alpha_x(i+1)} \quad \forall i \in [d-1], \quad \lambda_0 := 1 - x_{\alpha_x(1)}, \quad \lambda_d := x_{\alpha_x(d)}.$$

Recalling the definition of consistent permutation, we get

$$\sum_{i=0}^{d} \lambda_i = 1, \quad \lambda_i \geq 0, \quad \forall i \in \{0, \ldots, d\}$$

and $\tilde{f}(x)$ is a convex combination of $f\left(S^{x,0}\right), \ldots, f\left(S^{x,d}\right)$. In addition, we can calculate that

$$\left(\sum_{i=0}^{d} \lambda_i S^{x,i}\right)_{\alpha_x(k)} = \sum_{i=0}^{d} \lambda_i \cdot S_{\alpha_x(k)}^{x,i} = \sum_{i=0}^{d} \lambda_i \cdot \mathbf{1}(i \leq k) = \sum_{i=k}^{d} \lambda_i = x_{\alpha_x(k)},$$

which implies that

$$x = \sum_{i=0}^{d} \lambda_i S^{x,i}.$$

If $x^* \notin \left\{S^{x,0}, \ldots, S^{x,d}\right\}$, the assumption that the sub-optimality gap is at least $c$ gives

$$\tilde{f}(x) - f^* = \sum_{i=0}^{d} \lambda_i \left[f\left(S^{x,i}\right) - f^*\right] \geq \sum_{i=0}^{d} \lambda_i \cdot c = c.$$

Combining with $\|x - x^*\|_\infty \leq 1$, we have

$$\|x - x^*\|_\infty \leq c^{-1} \cdot \left[\tilde{f}(x) - f^*\right].$$

Otherwise if $x^* = S^{x,i}$ for some $i \in \{0, \ldots, d\}$. Then, we know

$$\tilde{f}(x) - f^* = \sum_{i=0}^{d} \lambda_i \left[f\left(S^{x,i}\right) - f^*\right] \geq \sum_{i \neq k} \lambda_i \cdot c = (1 - \lambda_k)c$$

and

$$\|x - x^*\|_\infty = \left\|\sum_{i=0}^{d} \lambda_i S^{x,i} - x^*\right\|_\infty = \left\|\sum_{i=0}^{d} \lambda_i \left(S^{x,i} - x^*\right)\right\|_\infty = \left\|\sum_{i \neq k} \lambda_i \left(S^{x,i} - x^*\right)\right\|_\infty$$

$$\leq \sum_{i \neq k} \lambda_i \left\|S^{x,i} - x^*\right\|_\infty \leq \sum_{i \neq k} \lambda_i = 1 - \lambda_k,$$

where the last inequality is because $S^{x,i}$ and $x^*$ are in the same hypercube $\mathcal{C}_y$. Combining the above two inequalities, it follows that

$$\|x - x^*\|_2 \leq c^{-1} \cdot \left[\tilde{f}(x) - f^*\right],$$

---
**Algorithm 3** Adaptive truncated SSGD method for the PAC guarantee
---
**Input:** Model $\mathcal{X}, \mathcal{B}_\mathsf{Y}, F(x, \xi_x)$, optimality guarantee parameter $\delta$, indifference zone parameter $c$.
**Output:** An $(c, \delta)$-PAC solution $x^*$ to problem (1).
 1: Set the initial guarantee $\epsilon_0 \leftarrow cN/4$.
 2: Set the number of epochs $E \leftarrow \lceil \log_2(N) \rceil + 1$.
 3: Set the initial search space $\mathcal{Y}_0 \leftarrow [1, N]^d$.
 4: **for** $e = 0, \ldots, E - 1$ **do**
 5:     Use Algorithm 2 to get an $(\epsilon_e, \delta/(2E))$-PAC solution $x_e$ in $\mathcal{Y}_e$.
 6:     Update guarantee $\epsilon_{e+1} \leftarrow \epsilon_e/2$.
 7:     Update the search space $\mathcal{Y}_{e+1} \leftarrow \mathcal{N}(x_e, 2^{-e-2}N)$.
 8: **end for**
 9: Round $x_{E-1}$ to an integral point satisfying the $(c, \delta)$-PAC guarantee by Algorithm 1.
---

which means that the WSM condition holds with $\kappa = c^{-1}$. Now we consider point $x \in [1, N]^d$ such that $\|x - x^*\|_\infty \geq 1$. We define

$$\tilde{x} := x^* + \frac{x - x^*}{\|x - x^*\|_\infty}$$

to be the point on the segment $\overline{xx^*}$ such that $\|\tilde{x} - x^*\|_\infty = 1$. By the convexity of $\tilde{f}(x)$ and the WSM condition for point $\tilde{x}$, we know

$$\tilde{f}(x) - f^* \geq \frac{\|x - x^*\|_\infty}{\|\tilde{x} - x^*\|_\infty} \left[ \tilde{f}(\tilde{x}) - f^* \right] = \frac{\tilde{f}(\tilde{x}) - f^*}{\|\tilde{x} - x^*\|_\infty} \cdot \|x - x^*\|_\infty \geq c^{-1} \cdot \|x - x^*\|_\infty,$$

which shows that the WSM condition holds with $\kappa = c^{-1}$. Hence, the WSM condition holds for all points in $[1, N]^d$ with $\kappa = c^{-1}$. $\qquad\square$

Using the WSM condition, we can accelerate Algorithm 2 by dynamically shrinking the search space. To describe the shrinkage of search space, we define the $\ell_\infty$-neighbourhood of point $x$ as

$$\mathcal{N}(x, a) := \{y \in [1, N]^d : \|y - x\|_\infty \leq a\}$$

and the orthogonal projection onto $\mathcal{N}(x, a)$ as

$$\mathcal{P}_{x,a}(y) := (y \wedge (x + a)\mathbf{1}) \vee (x - a)\mathbf{1}, \quad \forall x \in \mathbb{R}^d.$$

Now we give the adaptive truncated SSGD algorithm for the PAC guarantee. Basically, the algorithm finds a $(c, \delta)$-PAC solution and, with the assumption that the sub-optimality gap is at least $c$, the solution is the global optimum with probability at least $1 - \delta$. We prove that the computational complexity of Algorithm 3 has only $O(\log(N))$ dependence on $N$.

**Theorem 10.** *Suppose that Assumptions 1-4 hold. Then, Algorithm 3 returns a $(c, \delta)$-PAC solution. Furthermore, we have*

$$T(\delta, \mathcal{MC}_c) = O\left[ \frac{d^2 \log(N)}{c^2} \log(\frac{1}{\delta}) + \frac{d^3 \log(N)}{c^2} \log\left(\frac{d^2 N}{\epsilon^3}\right) + \frac{d^3 \log(N) L^2}{c^2} \right]$$

$$= \tilde{O}\left[ \frac{d^2 \log(N)}{c^2} \log(\frac{1}{\delta}) \right].$$

*Proof.* We first prove the correctness of Algorithm 3. Let $x^*$ be the minimizer of $f(x)$ and $f^* := f(x^*)$. We use the induction method to prove that, for each epoch $e$, it holds

$$\tilde{f}(x_e) - f^* \leq \epsilon_e$$

with probability at least $1 - (e + 1)\delta/(2E)$. For epoch 0, the solution $x_0$ is $(\epsilon_0, \delta/(2E))$-PAC and we know

$$\tilde{f}(x_0) - f^* \leq \epsilon_0$$

holds with probability at least $1 - \delta/(2E)$. We assume that the above event happens for the $(e - 1)$-th epoch with probability at least $1 - e \cdot \delta/(2E)$ and consider the case when this event happens. By

Theorem 9, function $\tilde{f}(x)$ satisfies the WSM condition with $\kappa = c^{-1}$. Hence, the intermediate solution $x_{e-1}$ satisfies

$$\|x_{e-1} - x^*\|_\infty \le c^{-1} \left[ \tilde{f}(x_{e-1}) - f^* \right] \le c^{-1} \epsilon_{e-1} = c^{-1} \cdot 2^{-e+1} \epsilon_0 = 2^{-e-1} N,$$

which implies that $x^* \in \mathcal{N}(x_{e-1}, 2^{-e-1}N) = \mathcal{N}_e$ and therefore $x^* \in \mathcal{N}_e$. For the epoch $e$, it holds

$$\tilde{f}(x_e) - f^* = \tilde{f}(x_e) - \min_{x \in \mathcal{N}_e} \tilde{f}(x) \le \epsilon_e$$

with probability at least $1 - \delta/(2E)$. Hence, the above event happens with probability at least $1 - \delta/(2E) - e \cdot \delta/(2E) = 1 - (e+1)\delta/(2E)$ for epoch $e$. By the induction method, we know the claim holds for all epochs. Considering the last epoch, we know

$$\tilde{f}(x_{E-1}) - f^* \le \epsilon_{E-1} = 2^{-E+1}\epsilon_0 = 2^{-\lceil \log_2(N) \rceil - 2} \cdot cN \le 2^{-\log_2(N)-2} \cdot cN = c/4$$

holds with probability at least $1 - \delta/2$. Thus, we know $x_{E-1}$ satisfies the $(c/4, \delta/2)$-PAC guarantee. By Theorem 2, the integral solution returned by Algorithm 3 satisfies the $(c/2, \delta)$-PAC guarantee. Since the indifference zone parameter is $c$, the solution satisfying the $(c/2, \delta)$-PAC guarantee must satisfies the $(c, \delta)$-PAC guarantee.

Next, we estimate the computational complexity of Algorithm 3. By Theorem 3, the computational complexity of epoch $e$ is at most

$$\tilde{O} \left[ \frac{d^2 (2^{-e}N)^2}{\epsilon_e^2} \log(\frac{E}{\delta}) \right] = \tilde{O} \left[ \frac{d^2 (2^{-e}N)^2}{(2^{-e-2} \cdot cN)^2} \log(\frac{E}{\delta}) \right] = \tilde{O} \left[ \frac{d^2}{c^2} \log(\frac{1}{\delta}) \right].$$

Summing over $e = 0, 1, \ldots, E-1$, we know the total computational complexity of $E$ epochs is at most

$$\tilde{O} \left[ E \cdot \frac{d^2}{c^2} \log(\frac{1}{\delta}) \right] = \tilde{O} \left[ \frac{d^2 \log(N)}{c^2} \log(\frac{1}{\delta}) \right].$$

By Theorem 2, the computational complexity of the rounding process is at most

$$\tilde{O} \left[ \frac{d}{c^2} \log(\frac{1}{\delta}) \right].$$

Combining the two parts, we know the computational complexity of Algorithm 3 is at most

$$\tilde{O} \left[ \frac{d^2 \log(N)}{c^2} \log(\frac{1}{\delta}) \right].$$

$\square$

Similarly, we can estimate the computational complexity under Assumption 5 and we omit the proof.

**Theorem 11.** *Suppose that Assumptions 1-5 hold. Then, Algorithm 3 returns a $(c, \delta)$-PAC solution. Furthermore, we have*

$$T(\delta, \mathcal{MC}_c) = \tilde{O} \left[ \frac{\beta(L+G)^2 \log(N) + d}{c^2} \log(\frac{1}{\delta}) \right].$$

# D Proofs in Section 4

## D.1 Proof of Theorem 5

The main idea of the generalization to multi-dimensional problems is to view optimization algorithms as estimators to the optimal value. This intuition is elaborated in the following definition.

**Definition 4.** Given a constant $C > 0$, we say that an algorithm is **sub-Gaussian with dimension $d$ and parameter $C$** if for any $d$-dimensional $L^\natural$-convex problem, any $\epsilon > 0$ and small enough $\delta > 0$, the algorithm returns an estimate to $f^*$ satisfying $|\hat{f}^* - f^*| \le \epsilon$ with probability at least $1 - \delta$ using at most

$$T(\epsilon, \delta) = \tilde{O} \left[ \frac{2C}{\epsilon^2} \log(\frac{2}{\delta}) \right]$$

objective function value evaluations.

We note that if we treat algorithms as estimators, estimators are generally "biased". This fact implies that the empirical mean of several $(\epsilon, \delta)$-PAC solutions does not produce a better optimality guarantee, while the empirical mean of several unbiased estimators usually has a tighter deviation bound.

Now, we inductively construct sub-Gaussian algorithms for problem (1). We first define the marginal objective function as

$$f^{d-1}(x) := \min_{y \in [N]^{d-1}} f(y, x). \tag{21}$$

Observe that each evaluation of $f^{d-1}(x)$ requires solving a $(d-1)$-dimensional $L^\natural$-convex sub-problem. Hence, if we have an algorithm for $(d-1)$-dimensional $L^\natural$-convex problems, we only need to solve the last one-dimensional problem

$$\min_{x \in [N]} f^{d-1}(x) = \min_{x \in [N]} \min_{y \in [N]^{d-1}} f(y, x) = \min_{x \in \mathcal{X}} f(x) \tag{22}$$

Moreover, we can prove that problem (22) is also a convex problem.

**Lemma 10.** *If function $f(x)$ is $L^\natural$-convex, then function $f^{d-1}(x)$ is $L^\natural$-convex on $[N]$.*

*Proof.* Let $k \in \{2, 3, \ldots, N-1\}$. By the definition of $f^{d-1}(x)$, there exists vectors $y_{k-1}, y_{k+1} \in [N]^{d-1}$ such that

$$f^{d-1}(k-1) = f(y_{k-1}, k-1), \quad f^{d-1}(k+1) = f(y_{k+1}, k+1).$$

By the $L^\natural$-convexity of $f(x)$, we have

$$
\begin{aligned}
f^{d-1}(k-1) + f^{d-1}(k+1) &= f(y_{k-1}, k-1) + f(y_{k+1}, k+1) \\
&\geq f\left(\left\lceil \frac{y_{k-1} + y_{k+1}}{2} \right\rceil, k\right) + f\left(\left\lfloor \frac{y_{k-1} + y_{k+1}}{2} \right\rfloor, k\right) \\
&\geq 2 \min_{y \in [N]^{d-1}} f(y, k) = 2 f^{d-1}(k),
\end{aligned}
$$

which means the discrete midpoint convexity holds at point $k$. Since we can choose $k$ arbitrarily, we know function $f^{d-1}(x)$ is convex on $[N]$. $\square$

Based on the observations above, we can use sub-Gaussian algorithms for $(d-1)$-dimensional problems and the uniform sampling algorithm to construct sub-Gaussian algorithms for $d$-dimensional problems. We give the pseudo-code in Algorithm 4. We prove that Algorithm 4 is sub-Gaussian with dimension $d$ and estimate its parameter.

**Theorem 12.** *Suppose that Assumptions 1-3 hold, and that Algorithm $\mathcal{A}$ is sub-Gaussian with dimension $d-1$ and parameter $C$. Then, Algorithm 4 is a sub-Gaussian algorithm with dimension $d$ and parameter $MC$, where $M > 0$ is an absolute constant.*

*Proof.* We first verify the correctness of Algorithm 4. The algorithm is the same as the uniform sampling algorithm in Zhang et al. (2021) except the condition for implementing Type-II Operations. Hence, if we can prove that, when Type-II Operations are implemented, it holds

$$h \leq |\mathcal{S}| \cdot \epsilon/80, \tag{23}$$

then the proof of Theorem 2 in Zhang et al. (2021) can be directly applied to this case. If the confidence interval is updated at the beginning of current iteration, then we have

$$h = |\mathcal{S}| \cdot \epsilon/160 < |\mathcal{S}| \cdot \epsilon/80.$$

Otherwise, if the confidence interval is not updated in the current iteration. Then, we have $|\mathcal{S}| > N_{cur}/2$ and therefore

$$h = N_{cur} \cdot \epsilon/160 < 2|\mathcal{S}| \cdot \epsilon/160 = |\mathcal{S}| \cdot \epsilon/80.$$

Combining the two cases, we have inequality (23) and the correctness of Algorithm 4.

Next, we estimate the computational complexity of Algorithm 4. Denote the active sets when we update the confidence interval as $\mathcal{S}_1, \ldots, \mathcal{S}_m$, where $m \geq 1$ is the number of times when the

---

**Algorithm 4** Multi-dimensional uniform sampling algorithm

---

**Input:** Model $\mathcal{X}, \mathcal{B}_x, F(x, \xi_x)$, optimality guarantee parameters $\epsilon$ and $\delta$, sub-Gaussian algorithm $\mathcal{A}$ with dimension $d-1$.

**Output:** An $(\epsilon, \delta)$-PAC solution $x^*$ to problem (1).

1: Set the active set $\mathcal{S} \leftarrow [N]$.
2: Set the step size $d \leftarrow 1$ and the maximal number of comparisons $T_{max} \leftarrow N$.
3: Set $N_{cur} \leftarrow +\infty$.
4: **while** the size of $\mathcal{S}$ is at least 3 **do**          $\triangleright$ Iterate until $\mathcal{S}$ has at most 2 points.
5:     **if** $|\mathcal{S}| \leq N_{cur}/2$ **then**          $\triangleright$ Update the confidence interval.
6:         Record current active set size $N_{cur} \leftarrow |\mathcal{S}|$.
7:         Set the confidence width $h \leftarrow N_{cur} \cdot \epsilon/160$.
8:         For each $x \in \mathcal{S}$, use algorithm $\mathcal{A}$ to get an estimate to $f^{d-1}(x)$ such that

$$\left| \hat{f}^{d-1}(x) - f^{d-1}(x) \right| \leq h$$

        holds with probability at least $1 - \delta/(2T_{max})$.
9:     **end if**
10:     **if** $\hat{f}^{d-1}(x) + h \leq \hat{f}^{d-1}(y) - h$ for some $x, y \in \mathcal{S}$ **then**          $\triangleright$ **Type-I Operation**
11:         **if** $x < y$ **then**
12:             Remove all points $z \in \mathcal{S}$ with the property $z \geq y$ from $\mathcal{S}$.
13:         **else**
14:             Remove all points $z \in \mathcal{S}$ with the property $z \leq y$ from $\mathcal{S}$.
15:         **end if**
16:     **else**          $\triangleright$ **Type-II Operation**
17:         Update the step size $d \leftarrow 2d$.
18:         Update $\mathcal{S} \leftarrow \{x_{min}, x_{min} + d, \ldots, x_{min} + kd\}$, where $x_{min} = \min_{x \in \mathcal{S}} x$ and $k = \lceil |\mathcal{S}|/2 \rceil - 1$.
19:     **end if**
20: **end while**          $\triangleright$ Now $\mathcal{S}$ has at most 2 points.
21: For each $x \in \mathcal{S}$, use Algorithm $\mathcal{A}$ to obtain an estimate to $f^{d-1}(x)$ such that

$$\left| \hat{f}^{d-1}(x) - f^{d-1}(x) \right| \leq \epsilon/4$$

holds with probability at least $1 - \delta/(2T_{max})$.
22: Return $x^* \leftarrow \arg\min_{x \in \mathcal{S}} \hat{f}^{d-1}(x)$.

---

confidence interval is updated. Then, we know $|\mathcal{S}_1| = N$ and $|\mathcal{S}_m| \geq 3$. By the condition for updating the confidence interval, it holds

$$|\mathcal{S}_{k+1}| \leq |\mathcal{S}_k| \quad \forall k \in [m-1],$$

which implies

$$|\mathcal{S}_k| \geq 2^{m-k}|\mathcal{S}_m| \geq 3 \cdot 2^{m-k} \quad \forall k \in [m].$$

Since the algorithm $\mathcal{A}$ is sub-Gaussian with parameter $C$, for each $x \in \mathcal{S}_k$, the computational complexity for generating $\hat{f}^{d-1}(x)$ is at most

$$\frac{2C}{h^2} \log(\frac{2T_{max}}{\delta}) = \frac{2C}{160^{-2}|\mathcal{S}_k|^2\epsilon^2} \log(\frac{2T_{max}}{\delta}) = |\mathcal{S}_k|^{-2} \cdot \frac{51200C}{\epsilon^2} \log(\frac{2T_{max}}{\delta}).$$

Hence, the total computational complexity for the $k$-th update of confidence intervals is at most

$$|\mathcal{S}_k| \cdot |\mathcal{S}_k|^{-2} \cdot \frac{51200C}{\epsilon^2} \log(\frac{2T_{max}}{\delta}) = |\mathcal{S}_k|^{-1} \cdot \frac{51200C}{\epsilon^2} \log(\frac{2T_{max}}{\delta}) \leq 2^{k-m}/3 \cdot \frac{51200C}{\epsilon^2} \log(\frac{2T_{max}}{\delta}).$$

Summing over all iterations, we have the computational complexity of all iterations of Algorithm 4 is at most

$$\sum_{k=1}^{m} 2^{k-m}/3 \cdot \frac{51200C}{\epsilon^2} \log(\frac{2T_{max}}{\delta}) = (2 - 2^{1-m}) \cdot \frac{51200C}{3\epsilon^2} \log(\frac{2T_{max}}{\delta}) < \frac{102400C}{3\epsilon^2} \log(\frac{2T_{max}}{\delta}).$$

Now we consider the computational complexity of the last subproblem. Since the algorithm 4 is sub-Gaussian with parameter $C$, the computational complexity of the subproblem is at most

$$2 \cdot \frac{2C}{(\epsilon/4)^2} \log(\frac{2T_{max}}{\delta}) = \frac{64C}{\epsilon^2} \log(\frac{2T_{max}}{\delta}).$$

Hence, the total computational complexity of Algorithm 4 is at most

$$\frac{102400C}{3\epsilon^2} \log(\frac{2T_{max}}{\delta}) + \frac{64C}{\epsilon^2} \log(\frac{2T_{max}}{\delta}) < 17099 \cdot \frac{2C}{\epsilon^2} \log(\frac{2T_{max}}{\delta}).$$

When $\delta$ is small enough, we can choose $M = 17100$ and the computational complexity of Algorithm 4 is at most

$$\frac{2MC}{\epsilon^2} \log(\frac{2T_{max}}{\delta}),$$

which implies that Algorithm 4 is sub-Gaussian with dimension $d$ and parameter $MC$. $\qquad\square$

If we treat the noisy evaluation oracle $F(x, \xi_x)$ as a sub-Gaussian algorithm with dimension 0 and parameter $\sigma^2$, then Theorem 12 implies that there exists a sub-Gaussian algorithm with dimension 1 and parameter $\sigma^2 M$. However, the parameter $C$ of the uniform sampling algorithm is usually smaller than $\sigma^2 M$ and therefore the uniform sampling algorithm is preferred in the one-dimensional case. Using the results of Theorem 12 and the fact that the uniform sampling algorithm is sub-Gaussian with dimension 1, we can inductively construct sub-Gaussian algorithms with any dimension $d$ and this finishes the proof of Theorem 5.

### D.2 Proof of Theorem 6

As a counterpart of separation oracles, we introduce the stochastic separation oracle, namely the $(\epsilon, \delta)$-separation oracle, to characterize the accuracy of separation oracles in the stochastic case.

**Definition 5.** An $(\epsilon, \delta)$-**separation oracle** $((\epsilon, \delta)$-$\mathcal{SO})$ is a function on $[1, N]^d$ with the property that for any input $x \in [1, N]^d$, it outputs a vector $\hat{g} \in \mathbb{R}^d$ such that the inequality

$$f(y) \geq f(x) - \epsilon$$

holds with probability at least $1 - \delta$ for any $y \in [1, N]^d \cap H$, where the half space $H$ is defined as $\{z : \langle \hat{g}, z - x \rangle \geq 0\}$.

We give a concrete example of $(\epsilon, \delta)$-$\mathcal{SO}$ oracle and provide an upper bound on the computational complexity of evaluating each oracle. We define the averaged subgradient estimator as

$$\hat{g}^n_{\alpha_x(i)} := \hat{F}_n\left(S^{x,i}\right) - \hat{F}_n\left(S^{x,i-1}\right), \quad \forall i \in [d], \tag{24}$$

where $\alpha_x$ is a consistent permutation of $x$, $n \geq 1$ is the number of samples, and $\hat{F}_n$ is the empirical mean of $n$ independent evaluations of $F$. The following lemma gives a lower bound on $n$ to guarantee that $\hat{g}^n$ is an $(\epsilon, \delta)$-$\mathcal{SO}$ oracle.

**Lemma 11.** *Suppose that Assumptions 1-3 hold. If we choose*

$$n = \tilde{O}\left[\frac{dN^2}{\epsilon^2} \log\left(\frac{1}{\delta}\right)\right],$$

*then $\hat{g}^n$ is an $(\epsilon, \delta)$-$\mathcal{SO}$ oracle. Moreover, the expected computational complexity of generating an $(\epsilon, \delta)$-$\mathcal{SO}$ oracle is at most*

$$O\left[\frac{d^2N^2}{\epsilon^2} \log\left(\frac{1}{\delta}\right) + d\right] = \tilde{O}\left[\frac{d^2N^2}{\epsilon^2} \log\left(\frac{1}{\delta}\right)\right].$$

*Proof.* By the assumption that $F(x, \xi_x) - f(x)$ is sub-Gaussian with parameter $\sigma^2$ for any $x$, we know that $\hat{g}_{\alpha_x(i)} - g_{\alpha_x(i)}$ is the difference of two independent sub-Gaussian random variables and therefore

$$\hat{g}_{\alpha_x(i)} - g_{\alpha_x(i)} \sim \text{subGaussian}\left(2\sigma^2\right) \quad \forall i \in [d],$$

where $g$ is the subgradient of $f(x)$ defined in (5). Then, using the properties of sub-Gaussian random variables, it holds that

$$\hat{g}^n_{\alpha_x(i)} - g_{\alpha_x(i)} \sim \text{subGaussian}\left(\frac{2\sigma^2}{n}\right) \quad \forall i \in [d].$$

Recalling that components of $\hat{g}^n$ are mutually independent, we know

$$\langle \hat{g}^n - g, y - x \rangle = \sum_i \left(\hat{g}^n_{\alpha_x(i)} - g_{\alpha_x(i)}\right) \cdot (y - x)_{\alpha_x(i)} \sim \text{subGaussian}\left(\frac{2\sigma^2}{n} \cdot \|y - x\|_2^2\right).$$

Since $\|y - x\|_2^2 \le dN^2$, we know

$$\langle \hat{g}^n - g, y - x \rangle \sim \text{subGaussian}\left(\frac{2dN^2\sigma^2}{n}\right).$$

By the Hoeffding bound, it holds

$$|\langle \hat{g}^n - g, y - x \rangle| \le \sqrt{\frac{4dN^2\sigma^2}{n}\log\left(\frac{2}{\delta}\right)}$$

with probability at least $1 - \delta$. If we choose

$$n = \left\lceil \frac{4dN^2\sigma^2}{\epsilon^2}\log\left(\frac{2}{\delta}\right)\right\rceil \le \frac{4dN^2\sigma^2}{\epsilon^2}\log\left(\frac{2}{\delta}\right) + 1,$$

it follows that

$$|\langle \hat{g}^n - g, y - x \rangle| \le \epsilon. \tag{25}$$

Since $f(x)$ is a convex function and $g$ is a subgradient at point $x$, we have $f(y) \ge f(x) + \langle g, y - x \rangle$ for all $y \in [1, N]^d$. Combining with inequality (25) gives

$$f(y) \ge f(x) + \langle \hat{g}^n, y - x \rangle + \langle g - \hat{g}^n, y - x \rangle \ge f(x) + \langle \hat{g}^n, y - x \rangle - \epsilon \quad \forall y \in [1, N]^d$$

holds with probability at least $1 - \delta$. Then, considering the half space $H = \{y : \langle \hat{g}^n, y - x \rangle \le 0\}$, it holds

$$f(y) \ge f(x) + \langle \hat{g}^n, y - x \rangle - \epsilon \ge f(x) - \epsilon \quad \forall y \in [1, N]^d \cap H^c$$

with the same probability. Taking the minimum over $[1, N]^d \cap H^c$, it follows that the averaged stochastic subgradient provides an $(\epsilon, \delta)$-$\mathcal{SO}$ oracle. Finally, the expected computational complexity of each oracle evaluation is at most

$$d \cdot n \le \frac{4d^2N^2\sigma^2}{\epsilon^2}\log\left(\frac{2}{\delta}\right) + d = \tilde{O}\left[\frac{d^2N^2\sigma^2}{\epsilon^2}\log\left(\frac{1}{\delta}\right)\right].$$

$\square$

Using the above results, the cutting-plane methods in Vaidya (1996); Bertsimas & Vempala (2004) can be used to shrink the search polytope without excluding all $\epsilon$-approximate solutions with high probability. We give the pseudo-code of the dimension reduction algorithm in Algorithm 5.

*Proof of Theorem 6.* We first verify the correctness of Algorithm 5. If the optimal solution has been removed during the dimension reduction process, we claim that the optimal solutions are removed from the search set by some cutting plane. This is because the dimension reduction steps will not remove integral points from the current search set (Jiang, 2020). This implies that $x$ is not in the half space

$$H := \{y : \hat{g}^T y \le \hat{g}^T z\},$$

where $\hat{g}$ is an $(\epsilon/8, \delta/4)$-$\mathcal{SO}$ oracle. Then, by the definition of $(\epsilon/8, \delta/4)$-$\mathcal{SO}$ oracle and the claim that $x \in [1, N]^d \cap H^c$, it holds

$$\min_{x \in \mathcal{S}} f(x) \le \min_{x \in \mathcal{X}} f(x) + \epsilon/4 \tag{26}$$

with probability at least $1 - \delta/4$.

---

**Algorithm 5** Dimension reduction method for PAC guarantee

---

**Input:** Model $\mathcal{X}, \mathcal{B}_x, F(x, \xi_x)$, optimality guarantee parameters $\epsilon$ and $\delta$, $(\epsilon, \delta)$-$\mathcal{SO}$ oracle $\hat{g}$.
**Output:** An $(\epsilon, \delta)$-PAC solution $x^*$ to problem (1).
 1: Set the initial polytope $P \leftarrow [1, N]^d$.
 2: Initialize the set of points used to query separation oracles $\mathcal{S} \leftarrow \emptyset$.
 3: **for** $d' = d, d-1, \ldots, 2$ **do**
 4:     Apply Vaidya's method or the random walk-based cutting-plane method with $(\epsilon/4, \delta/4)$-$\mathcal{SO}$ oracles.
 5:     Add points to $\mathcal{S}$ whenever separation oracles are called.
 6:     Stop when the volume of $P$ is small enough.
 7:     Find the hyperplane $H$ that contains all integral points in $P$.
                          ▷ If $P$ contains no integral points, then an arbitrary hyperplane works.
 8:     Update $P$ to a polytope on the hyperplane $H$.          ▷ Reduce the dimension by 1.
 9: **end for**
10: Find an $(\epsilon/4, \delta/4)$-PAC solution of the last one-dim problem and add the solution to $\mathcal{S}$.
11: Find the $(\epsilon/4, \delta/4)$-PAC solution $\hat{x}$ of problem $\min_{x \in \mathcal{S}} f(x)$.
12: Round $\hat{x}$ to an integral solution by Algorithm 1.

---

Otherwise if the optimal solution has not been removed from the search set throughout the dimension reduction process, we know the last one-dimensional problem contains the optimal solution. Hence, the $(\epsilon/4, \delta/4)$-PAC solution to the one-dimensional problem is also an $(\epsilon/4, \delta/4)$-PAC solution to the original problem. Since the PAC solution is also added to the set $\mathcal{S}$, we also have relation (26) holds with probability at least $1 - \delta/4$. Then, the $(\epsilon/4, \delta/4)$-PAC solution $\bar{x}$ to problem $\min_{x \in \mathcal{S}} f(x)$ satisfies

$$f(\bar{x}) \le \min_{x \in \mathcal{X}} f(x) + \epsilon/2$$

with probability at least $1 - \delta/2$, or equivalently $\bar{x}$ is an $(\epsilon/2, \delta/2)$-PAC solution to problem (1). Using the results of Theorem 2, the solution returned by Algorithm 5 is an $(\epsilon, \delta)$-PAC solution.

Next, we estimate the expected computational complexity of Algorithm 5. By the results in Jiang (2020), $(\epsilon/4, \delta/4)$-$\mathcal{SO}$ oracles are called at most $O[d(d + \log(N))]$ times. Hence, the size of $\mathcal{S}$ is at most $O[d(d + \log(N))]$. By the estimates in Lemma 11, the total computational complexity of the dimension reduction process is at most

$$O\left[\frac{d^3 N^2 (d + \log(N))}{\epsilon^2} \log(\frac{1}{\delta}) + d^2(d + \log(N))\right] = \tilde{O}\left[\frac{d^3 N^2 (d + \log(N))}{\epsilon^2} \log(\frac{1}{\delta})\right].$$

Moreover, the one-dimensional convex problem has at most $N$ feasible points and Theorem 2 in Zhang et al. (2021) implies that the expected computational complexity for this problem is at most

$$\tilde{O}\left[\frac{1}{\epsilon^2} \log(\frac{1}{\delta})\right].$$

Since the size of $\mathcal{S}$ is at most $O[d(d + \log(N))]$, the sub-problem for the set $\mathcal{S}$ takes at most

$$O\left[\frac{d^2(d + \log(N))}{\epsilon^2} \log(\frac{1}{\delta}) + d^2(d + \log(N))\right] = \tilde{O}\left[\frac{d^2(d + \log(N))}{\epsilon^2} \log(\frac{1}{\delta})\right]$$

objective function value evaluations. Finally, Theorem 2 shows that the expected computational complexity of the rounding process is at most

$$\tilde{O}\left[\frac{d}{\epsilon^2} \log(\frac{1}{\delta})\right].$$

In summary, the total expected computational complexity of Algorithm 5 is at most

$$\tilde{O}\left[\frac{d^3 N^2 (d + \log(N))}{\epsilon^2} \log(\frac{1}{\delta})\right].$$

$\square$

# E  Proofs in Section 5

## E.1  Proof of Theorem 7

The information-theoretical inequality in Kaufmann et al. (2016) provides a systematic way to prove lower bounds of zeroth-order algorithms. Given an algorithm and a model $M$, we denote $N_x(\tau)$ as the number of times that $F(x, \xi_x)$ is evaluated when the algorithm terminates, where $\tau$ is the stopping time of the algorithm. Then, it follows from the definition that

$$\mathbb{E}_M[\tau] = \sum_{x \in \mathcal{X}} \mathbb{E}_M\left[N_x(\tau)\right],$$

where $\mathbb{E}_M$ is the expectation when the model $M$ is given. Similarly, we can define $\mathbb{P}_M$ as the probability when the model $M$ is given. The following lemma was proved in Kaufmann et al. (2016) and is the major tool for deriving lower bounds in this paper.

**Lemma 12.** *For any two models $M_1$, $M_2$ and any event $\mathcal{E} \in \mathcal{F}_\tau$, we have*

$$\sum_{x \in \mathcal{X}} \mathbb{E}_{M_1}\left[N_x(\tau)\right] \mathrm{KL}(\nu_{1,x}, \nu_{2,x}) \geq d(\mathbb{P}_{M_1}(\mathcal{E}), \mathbb{P}_{M_2}(\mathcal{E})), \tag{27}$$

*where $d(x, y) := x \log(x/y) + (1-x) \log((1-x)/(1-y))$, $\mathrm{KL}(\cdot, \cdot)$ is the KL divergence and $\nu_{k,x}$ is the distribution of model $M_k$ at point $x$ for $k = 1, 2$.*

The information-theoretical inequality (27) is our major tool for deriving lower bounds. We first reduce the construction of $L^\natural$-convex functions to the construction of submodular functions. Then, using the family of submodular functions defined in Graur et al. (2020), we can construct $d + 1$ submodular functions that have different optimal solutions and have the same value except on $d + 1$ potential solutions. Hence, the algorithm has to simulate enough samples on the $d + 1$ potential solutions to decide the optimal solution and the computational complexity is proportional to $d$.

*Proof of Theorem 7.* In this proof, we change the feasible set to $\mathcal{X} = \{0, 1, \ldots, N\}^d$, where $N \geq 1$. We split the proof into three steps.

**Step 1.**  We first show that the construction of $L^\natural$-convex functions can be reduced to the construction of submodular functions. Equivalently, we show that any submodular function defined on $\{0, 1\}^d$ can be extended to an $L^\natural$-convex function on $\mathcal{X}$ with the same convex extension after scaling. Let $g(x)$ be a submodular function defined on $\{0, 1\}^d$ and $\tilde{g}(x)$ be the Lovász extension of $g(x)$. We first extend the domain of the Lovász extension to $[0, N]^d$ by scaling, i.e.,

$$\tilde{f}(x) := \tilde{g}(x/N) \quad \forall x \in [0, N]^d.$$

Then, we define the discretization of $\tilde{f}(x)$ by restricting to the integer lattice

$$f(x) := \tilde{f}(x) \quad \forall x \in \mathcal{X}.$$

We prove that $f(x)$ is an $L^\natural$-convex function. By Proposition 7.25 in Murota (2003), we know the Lovász extension $\tilde{g}(x)$ is a polyhedral $L$-convex function. Since the scaling operation does not change the $L$-convexity, we know $\tilde{f}(x)$ is also polyhedral $L$-convex. Hence, by Theorem 7.29 in Murota (2003), the function $\tilde{f}(x)$ satisfies the $\mathrm{SBF}^\natural[\mathbb{R}]$ property, namely,

$$\tilde{f}(p) + \tilde{f}(q) \geq \tilde{f}[(p - \alpha\mathbf{1}) \vee q] + \tilde{f}(p \wedge (q + \alpha\mathbf{1})) \quad \forall p, q \in [0, N]^d, \ \alpha \geq 0.$$

Restricting to the integer lattice, we know the $\mathrm{SBF}^\natural[\mathbb{Z}]$ property holds for $f(x)$, namely,

$$f(p) + f(q) \geq f[(p - \alpha\mathbf{1}) \vee q] + f(p \wedge (q + \alpha\mathbf{1})) \quad \forall p, q \in \{0, \ldots, N\}^d, \ \alpha \in \mathbb{N}.$$

Finally, Theorem 7.7 in Murota (2003) shows that the $L^\natural$-convexity is equivalent to the $\mathrm{SBF}^\natural[\mathbb{Z}]$ property and therefore we know that $f(x)$ is an $L^\natural$-convex function.

**Step 2.** Next, we construct $d+1$ submodular functions on $\{0,1\}^d$ and extend them to $\mathcal{X}$ by the process defined in Step 1. The construction is based on the family of submodular functions defined in Graur et al. (2020). We denote $\mathcal{I} := \{0\} \cup [d]$. For each $i \in \mathcal{I}$, we define point $x^i \in \{0,1\}^d$ as

$$x^i := \sum_{j=1}^{i} e_j,$$

where $e_j$ is the $j$-th unit vector of $\mathbb{R}^d$. Index $j(x)$ is defined as the maximal index $j$ such that

$$x_i = 1 \quad \forall i \in [j].$$

If $x_1 = 0$, then we define $j(x) = 0$. Given $c : \mathcal{I} \mapsto \mathbb{R}$, we define a function on $\{0,1\}^d$ as

$$g^c(x) := \begin{cases} -c(i) & \text{if } x = x^i \text{ for some } i \in \mathcal{I} \\ (\|x\|_1 - j(x)) \cdot (d + 2 - \|x\|_1) & \text{otherwise.} \end{cases}$$

By Lemma 6 in Graur et al. (2020), the function $g^c(x)$ is submodular if $c(\mathcal{I}) \subset \{0,1\}$. Using the fact that convex combinations of submodular functions are still submodular, we know that $g^c(x)$ is submodular for any $c$ such that $c(\mathcal{I}) \subset [0,1]$. Then, for each $i \in \mathcal{I}$, we construct

$$c^i(0) := \frac{1}{2}, \quad c^i(j) := \begin{cases} 1 & j = i \\ 0 & j \neq i \end{cases} \quad \forall j \in [d].$$

We denote $g^i(x) := g^{c^i}(x)$ and let $f^i(x)$ be the extension of $6\epsilon \cdot g^i(x)$ on $\mathcal{X}$ by the process in Step 1. By the result in Step 1, we know that $f^i(x)$ is $L^\natural$-convex.

Next, we prove that $f^0(x)$ has disjoint set of $\epsilon$-optimal solutions with $f^i(x)$ for any $i \in [d]$. For each $f^i(x)$, we define the set of $\epsilon$-optimal solutions as

$$\mathcal{X}_\epsilon^i := \{x \in \mathcal{X} : f^i(x) - \min_y f^i(y) \leq \epsilon\}.$$

We first consider $\mathcal{X}_\epsilon^0$. By the definition of $g^0(x)$, we know that

$$f^0(x^0) = g^0(x^0) = -3\epsilon, \quad f^0(x) = g^0(x/N) \geq 0 \quad \forall x \in \{0,N\}^d \backslash \{x^0\}, \tag{28}$$

which implies that

$$\mathcal{X}_\epsilon^0 = \{x \in \mathcal{X} : f^0(x) \leq -2\epsilon\}.$$

Since $f^0(x)$ is defined by the scaled Lovász extension of $g^0(x)$, we have

$$f^0(x) = N^{-1} \cdot \left[ (N - x_{\alpha(1)})f^0(S^0) + \sum_{i=1}^{d-1} (x_{\alpha(i)} - x_{\alpha(i+1)})f^0(S^i) + x_{\alpha(d)}f^0(S^d) \right], \tag{29}$$

where $\alpha$ is a consistent permutation of $x/N$ and $S^i := N \cdot S^{x/N,i} \in \{0,N\}^d$ is the $i$-th neighbouring points of $x$ in the hypercube $\{0,N\}^d$. Using the relation in (28) and the fact $S^0 = x^0$, we get

$$f^0(x) \geq N^{-1} \cdot (N - x_{\alpha(1)})f(S_0) = N^{-1} \cdot (N - x_{\alpha(1)})f(x^0) = -3\epsilon N^{-1} \cdot (N - x_{\alpha(1)}).$$

Hence, for any point $x \in \mathcal{X}_\epsilon^0$, we have $N - x_{\alpha(1)} = N - \max_i x_i \geq 2N/3$ and therefore

$$\mathcal{X}_\epsilon^0 \subset \{x \in \mathcal{X} : N - \max_i x_i \geq 2N/3\} = \{x \in \mathcal{X} : \max_i x_i \leq N/3\}. \tag{30}$$

Next, we consider $\mathcal{X}_\epsilon^i$ with $i \in [d]$. By the definition of $g^i(x)$, we have

$$f^i(x^0) = g^i(x^0) = -3\epsilon, \quad f^i(x) = g^i(x) \geq -6\epsilon \quad \forall x \in \{0,N\}^d \backslash \{x^0\},$$

which implies that

$$\mathcal{X}_\epsilon^i = \{x \in \mathcal{X} : f^i(x) \leq -5\epsilon\}.$$

Since the consistent permutation and neighboring points only depend on the coordinate of $x$, we know

$$f^i(x) = N^{-1} \cdot \left[ (N - x_{\alpha(1)})f^i(S^0) + \sum_{i=1}^{d-1} (x_{\alpha(i)} - x_{\alpha(i+1)})f^i(S^i) + x_{\alpha(d)}f^i(S^d) \right] \tag{31}$$

$$\geq N^{-1} \cdot \left[ -3\epsilon(N - x_{\alpha(1)}) - 6\epsilon \sum_{i=1}^{d-1} (x_{\alpha(i)} - x_{\alpha(i+1)}) - 6\epsilon \cdot x_{\alpha(d)} \right]$$

$$= N^{-1} \cdot \left[ -3\epsilon(N - x_{\alpha(1)}) - 6\epsilon \cdot x_{\alpha(1)} \right] = -3\epsilon N^{-1} \cdot (N + x_{\alpha(1)}).$$

Hence, the set $\mathcal{X}_\epsilon^i$ satisfies

$$\mathcal{X}_\epsilon^i \subset \{x \in \mathcal{X} : N + \max_i x_i \geq 5N/3\} = \{x \in \mathcal{X} : \max_i x_i \geq 2N/3\}. \tag{32}$$

Combining the relations (30) and (32), we know $\mathcal{X}_\epsilon^0 \cap \mathcal{X}_\epsilon^i = \emptyset$ for all $i \in [d]$.

**Step 3.** Finally, we give a lower bound of $T_0(\epsilon, \delta, \mathcal{MC})$. For each $i \in \mathcal{I}$, we define $M_i$ as the model such that the objective function is $f^i(x)$ and the distribution at each point is Gaussian with variance $\sigma^2$. Same as the one-dimensional case, given a zeroth-order algorithm and a model $M$, we denote $N_x(\tau)$ as the number of times that $F(x, \xi_x)$ is simulated when the algorithm terminates. By definition, we have

$$\mathbb{E}_M[\tau] = \sum_{x \in \mathcal{X}} \mathbb{E}_M [N_x(\tau)],$$

where $\mathbb{E}_M$ is the expectation when the model $M$ is given. Similarly, we can define $\mathbb{P}_M$ as the probability when the model $M$ is given. Suppose $\mathcal{A}$ is an $[(\epsilon, \delta)\text{-PAC}, \mathcal{MC}]$-algorithm and let $\mathcal{E}$ be the event that the solution returned by $\mathcal{A}$ is in the set $\mathcal{X}_\epsilon^0$. Since $\mathcal{X}_\epsilon^0 \cap \mathcal{X}_\epsilon^i = \emptyset$ for all $i \in [d]$, we know

$$\mathbb{P}_{M_0}[\mathcal{E}] \geq 1 - \delta, \quad \mathbb{P}_{M_i}[\mathcal{E}] \leq \delta \quad \forall i \in [d].$$

Using the information-theoretical inequality (27), it holds

$$\sum_{x \in \mathcal{X}} \mathbb{E}_{M_0} [N_x(\tau)] \, \mathrm{KL}(\nu_{0,x}, \nu_{i,x}) \geq d(\mathbb{P}_{M_0}(\mathcal{E}), \mathbb{P}_{M_i}(\mathcal{E})) \geq d(1 - \delta, \delta) \geq \log(\frac{1}{2.4\delta}), \tag{33}$$

where $d(x, y) := x \log(x/y) + (1 - x) \log((1 - x)/(1 - y))$, $\mathrm{KL}(\cdot, \cdot)$ is the KL divergence and $\nu_{i,x}$ is the distribution of $F^i(x, \xi_x)$. Since the distributions $\nu_{i,x}$ are Gaussian with variance $\sigma^2$, the KL divergence can be calculated as

$$\mathrm{KL}(\nu_{0,x}, \nu_{i,x}) = 2\sigma^{-2} \left( f^0(x) - f^i(x) \right)^2.$$

Now we estimate $f^0(x) - f^i(x)$ for all $i \in [d]$. By equations (29) and (31), we get

$$f^0(x) - f^i(x) = N^{-1} \left[ (N - x_{\alpha(1)}) \left( f^0(S^0) - f^i(S^0) \right) \right. \tag{34}$$

$$\left. + \sum_{j=1}^{d-1} (x_{\alpha(j)} - x_{\alpha(j+1)}) \left( f^0(S^j) - f^i(S^j) \right) + x_{\alpha(d)} \left( f^0(S^d) - f^i(S^d) \right) \right],$$

where $\alpha$ is a consistent permutation of $x/N$ and $S^i$ is the $i$-th neighboring point of $x$ in hypercube $\{0, N\}^d$. By the definition of $f^0(x)$ and $f^i(x)$, we have

$$f^0(x) - f^i(x) = \begin{cases} 6\epsilon & \text{if } x = x^i \\ 0 & \text{otherwise.} \end{cases}$$

Since $\left\| x^i \right\|_1 = i$ and $\left\| S^j \right\|_1 = j$ for all $j \in \mathcal{I}$, we know

$$f^0(S^i) - f^i(S^i) \leq 6\epsilon, \quad f^0(S^j) - f^i(S^j) = 0 \quad \forall j \in \mathcal{I} \backslash \{i\}.$$

Substituting into equation (34), it follows that

$$f^0(x) - f^i(x) \leq \begin{cases} \left( 6\epsilon \cdot (x_{\alpha(i)} - x_{\alpha(i+1)}) \right)^2 & \text{if } i \in [d-1] \\ \left( 6\epsilon \cdot x_{\alpha(d)} \right)^2 & \text{if } i = d. \end{cases}$$

Hence, the KL divergence is bounded by

$$\mathrm{KL}(\nu_{0,x}, \nu_{i,x}) = 2\sigma^{-2} \left( f^0(x) - f^i(x) \right)^2 \leq \begin{cases} 72\sigma^{-2} N^{-2} \epsilon^2 \left( (x_{\alpha(i)} - x_{\alpha(i+1)}) \right)^2 & \text{if } i \in [d-1] \\ 72\sigma^{-2} N^{-2} \epsilon^2 x_{\alpha(d)}^2 & \text{if } i = d. \end{cases}$$

Substituting the KL divergence into inequality (33) and summing over $i = 1, \ldots, d$, we get

$$\sum_{x \in \mathcal{X}} \mathbb{E}_{M_0} \left[ N_x(\tau) \right] \cdot 72 \sigma^{-2} N^{-2} \epsilon^2 \left[ \sum_{i=1}^{d-1} (x_{\alpha(i)} - x_{\alpha(i+1)})^2 + x_{\alpha(d)}^2 \right] \geq d \log \left( \frac{1}{2.4\delta} \right). \quad (35)$$

Since $\alpha$ is the consistent permutation of $x$, we know

$$0 \leq x_{\alpha(i)} - x_{\alpha(i+1)} \leq N \quad \forall i \in [d-1]$$

and therefore

$$\sum_{i=1}^{d-1} (x_{\alpha(i)} - x_{\alpha(i+1)})^2 + x_{\alpha(d)}^2 \leq N \cdot \left( \sum_{i=1}^{d-1} (x_{\alpha(i)} - x_{\alpha(i+1)}) + x_{\alpha(d)} \right) = N \cdot x_{\alpha(1)} \leq N^2.$$

Combining with inequality (35), we get

$$\sum_{x \in \mathcal{X}} \mathbb{E}_{M_0} \left[ N_x(\tau) \right] \cdot 72 \epsilon^2 \sigma^{-2} \geq d \log \left( \frac{1}{2.4\delta} \right),$$

which implies that

$$\mathbb{E}_{M_0}[\tau] = \sum_{x \in \mathcal{X}} \mathbb{E}_{M_0} \left[ N_x(\tau) \right] \geq \frac{d\sigma^2}{72\epsilon^2} \log \left( \frac{1}{2.4\delta} \right).$$

$\square$

# F   More numerical experiments

## F.1   Separable convex function minimization

We consider the problem of minimizing a stochastic function whose expectation is a separable $L^\natural$-convex function of the form

$$f_{c,x^*}(x) := \sum_{i=1}^{d} c_i g(x_i^*; x_i),$$

where $c_i \in [0.75, 1.25]$, $x_i^* \in \{1, \ldots, \lfloor 0.3N \rfloor\}$ for all $i \in [d]$ and

$$g(y^*; y) := \begin{cases} \sqrt{\frac{y^*}{y}} - 1 & \text{if } y \leq y^* \\ \sqrt{\frac{N+1-y^*}{N+1-y}} - 1 & \text{if } y > y^* \end{cases} \quad \forall y, y^* \in [N].$$

It can be observed that the function $f_{c,x^*}(x)$ is the sum of separable convex functions and therefore is $L^\natural$-convex. Moreover, the function $f_{c,x^*}(x)$ has the optimum $x^*$ associated with the optimal value 0. For stochastic evaluations, we add Gaussian noise with mean 0 and variance 1. Figure 1 in Zhang et al. (2021) shows that the landscape of $g(x^*; x)$ is very flat. The advantage of this numerical example is that the expected objective function has a closed form, and we are able to exactly compute the optimality gap of the solutions returned by the proposed algorithms.

The dimension and scale of the separable convex model are chosen as $d \in \{2, 6, 10\}$ and $N \in \{50, 100, 150\}$. The optimality guarantee parameters are chosen as $\epsilon = (d!)^{1/d}/5$ and $\delta = 10^{-6}$, respectively. The choice of $\epsilon$ ensures that the $\epsilon$-sub-level set of objective function covers a "flat region" of the landscape. Same as the one-dimensional case, we compute the average computational complexity of 400 independently generated models to approximate the expected computational complexity. Moreover, early stopping conditions are designed to terminate algorithms early when little progress is made at any iteration. For the truncated subgradient descent method, we maintain the empirical mean of stochastic objective function values up to the current iteration and terminate the algorithm if the empirical mean does not decrease by $\epsilon/\sqrt{N}$ after $O(d\epsilon^{-2} \log(1/\delta))$ consecutive iterations. For the dimension reduction method, we terminate the algorithm if the polytope is empty. Furthermore, we have observed that using $(N\epsilon/4, \delta/4)$-$\mathcal{SO}$ oracles is enough for producing high-probability guarantees within the range $1 \leq N \leq 150$.

### F.2 Optimal allocation problem of a service system

In this subsection, we implement our proposed algorithms on the optimal allocation problem of a queueing system. We consider the 24-hour operation of a service system with a single stream of incoming customers. The customers arrive according to a a doubly stochastic non-homogeneous Poisson process with intensity function

$$\Lambda(t) := 0.5\lambda N \cdot (1 - |t - 12|/12) \quad \forall t \in [0, 24],$$

where $\lambda$ is a positive constant and $N$ is a positive integer. Each customer requests a service with service time independent and identically distributed according to the log-normal distribution with mean $1/\lambda$ and variance $0.1$. We divide the 24-hours operation into $d$ time slots with length $24/d$ for some positive integer $d$. For the $i$-th time slot, there are $x_i \in [N]$ of homogeneous servers that work independently in parallel and the number of servers cannot be changed during the slot. Assume that the system operates based on a first-come-first-serve routine, with unlimited waiting room in each queue, and that customers never abandon.

The decision maker's objective is to select the staffing level $x := (x_1, \ldots, x_d)$ such that the total waiting time of all customers is minimized. Namely, letting $f(x)$ be the expected total waiting time under the staffing plan $x$, then the optimization problem can be written as

$$\min_{x \in [N]^d} f(x). \tag{36}$$

It has been proved in Altman et al. (2003) that the function $f(\cdot)$ is multimodular. We define the linear transformation

$$g(y) := (y_1, y_2 - y_1, \ldots, y_d - y_{d-1}) \quad \forall y \in \mathbb{R}^d.$$

Then, Murota (2003) has proved that

$$h(y) := f \circ g(y) = f(y_1, y_2 - y_1, \ldots, y_d - y_{d-1})$$

is an $L^\natural$-convex function on the $L^\natural$-convex set

$$\mathcal{Y} := \{y \in [Nd]^d \mid y_1 \in [N], \ y_{i+1} - y_i \in [N], \ i = 1, \ldots, N - 1\}.$$

The optimization problem (36) has the trivial solution $x_1 = \cdots = x_d = N$. However, in reality, it is also necessary to keeping the staffing cost low. There are two different approaches to achieve this goal. Firstly, we can constrain the total number of servers $\sum_{i=1}^d x_i$ to be at most $K$, where $K \leq Nd$ is a positive integer and the optimization problem can be written as

$$\min_{y \in \mathcal{Y}} h(y) \quad \text{s.t. } y_d \leq K. \tag{37}$$

On the other hand, we can add a regularization term $R(x_1, \ldots, x_d) := c/d \cdot \sum_{i=1}^d x_i = c/d \cdot y_d$ to the objective function, where $c$ is a positive constant. The optimization problem can be written as

$$\min_{y \in \mathcal{Y}} h(y) + c/d \cdot y_d. \tag{38}$$

We refer problems (37) and (38) as the constrained and the regularized problems, respectively. Our algorithms can be extended to this case by considering the Lovász extension $\tilde{h}(y)$ on the set

$$\tilde{\mathcal{Y}} := \{y \in [1, Nd]^d \mid y_1 \in [1, N], \ y_{i+1} - y_i \in [1, N], \ i = 1, \ldots, N - 1\}.$$

We compare the performance of the projected SSGD method (Algorithm 2) with truncation ($M < \infty$) and without truncation ($M = \infty$) on both problems. In the truncation-free case, the step size is chosen to be $\eta = O(N\sqrt{d/T})$. We first fix the dimension (number of time slots) to be $d = 4$ and compare the performance when the scale $N \in \{10, 20, 30, 40, 50\}$, and we then fix the scale to be $N = 10$ and compare the performance when the dimension $d \in \{4, 8, 12, 16, 20, 24\}$. The parameters of the problem are chosen as $\lambda = 4$, $c = 50$ and $K = \lfloor Nd/3 \rfloor$, and the optimality guarantee parameters are $\epsilon = N/2$ and $\delta = 10^{-6}$. For each problem setup, we average the computational complexities of 10 independent implementations to estimate the expected computational complexity. Moreover, early stopping is used to terminate algorithms early when little progress is made after some iterations. More concretely, we maintain the empirical mean of stochastic objective function values up to the current iteration and terminate the algorithm if the empirical mean does not decrease by $\epsilon/\sqrt{N}$ after $O(d\epsilon^{-2} \log(1/\delta))$ consecutive iterations.

We first implement both algorithms on the trivial problem (36) for 10 times. Since the optimal solution is known, it is possible to verify whether the solutions returned by algorithms are at most $\epsilon$ worse than the optimum. It turns out that both algorithms succeed in implementations with any parameters $d$ and $N$. Next, we consider the performance of algorithms on problems (37) and (38), and the results are summarized in Table 3.