# OpenReview forum: "Stochastic $L^\natural$-convex Function Minimization"
_NeurIPS.cc/2021/Conference — NeurIPS 2021 Poster_

### Official Review · Reviewer_2nJo · 2021-07-04

**Rating:** 6
**Confidence:** 4

**Summary:**

This paper studies stochastic L# -convex minimization problem and develops the "polynomial" time algorithms that return a near-optimal solution.

**Limitations And Societal Impact:**

It is a theoretical work and potentially has a significant impact on the society.

**Main Review:**

1. Originality.
The algorithmic results on stochastic L# convex programs are new and important.

2. Quality and Significance.
Many results are simply applying existing ones, and the "polynomial-time" algorithm may be incorrect.
1). All the algorithms presented in this paper have running time proportional to N, an input of the stochastic L# convex programs. This means that the proposed algorithms are rather pseudo-polynomial. This is an important concept, since the Knapsack problem admits a pseudo-polynomial algorithm but is NP-complete. Please carefully define it and clarify it.
2). The paper is literally done by showing the Lovas extension (i.e., Theorem 1).
For example, Theorem 2 is due to convexity and Hoeffding bound.
Theorems 3 and 4 are anazlying the stochastic subgradient method. What is fundamental difference between this theorem and others in the existing literature? Such as

Boyd, Stephen, and Almir Mutapcic. "Stochastic subgradient methods." Lecture Notes for EE364b, Stanford University (2008).

Davis, Damek, et al. "Stochastic subgradient method converges on tame functions." Foundations of computational mathematics 20.1 (2020): 119-154.

Ruszczyński, Andrzej. "Convergence of a stochastic subgradient method with averaging for nonsmooth nonconvex constrained optimization." Optimization Letters (2020): 1-11.

Theorem 6 is a simple application of Jiang (2020) via LLL algorithm (Lenstra et al., 1982)

3). The lower bound result in Theorem 7 is interesting. Is there a polynomial-time algorithm for solving L# convex programs? At least we know that for [0,1} submodular minimization, it is a strongly polynomial-time solvable.

3. Clarity.
Algorithm 1: please define the neighboring points in the main paper.

**Time Spent Reviewing:**

8

---

> ### Author Response · Authors · 2021-08-09
> **Response to Reviewer 2nJo**
>
> We thank the reviewer for the detailed comments and suggestions. Please find our response to the comments below.
>
> 1. All the algorithms presented in this paper have running time proportional to N, an input of the stochastic L# convex programs. This means that the proposed algorithms are rather pseudo-polynomial. This is an important concept, since the Knapsack problem admits a pseudo-polynomial algorithm but is NP-complete. Please carefully define it and clarify it.
>
> Response: We thank the reviewer for the very helpful comment and we are sorry for the confusion. If the parameter N is treated as an input parameter, we agree with the reviewer that the proposed algorithms in this work are all pseudo-polynomial. That said, the message we hope to convey is that the sample complexities of the algorithms in Section 4 are upper bounded by a constant that is independent of the objective function. This property is important for some applications of the stochastic L# function minimization problem where the objective function has unfavorable properties (e.g. large Lipschitz constant, small sub-optimality gap). Such unfavorable properties may not be known a priori before the problem is solved.  Therefore, in this work, we did not treat the parameter N as an input parameter but focus on reducing the dependence on the objective function. We will make the definition more clear in the revised paper.
>
> 2. The paper is literally done by showing the Lovasz extension (i.e., Theorem 1). For example, Theorem 2 is due to convexity and Hoeffding bound. Theorems 3 and 4 are analyzing the stochastic subgradient method. What is fundamental difference between this theorem and others in the existing literature? Such as
>
> Boyd, Stephen, and Almir Mutapcic. "Stochastic subgradient methods." Lecture Notes for EE364b, Stanford University (2008).
>
> Davis, Damek, et al. "Stochastic subgradient method converges on tame functions." Foundations of computational mathematics 20.1 (2020): 119-154.
>
> Ruszczyński, Andrzej. "Convergence of a stochastic subgradient method with averaging for nonsmooth nonconvex constrained optimization." Optimization Letters (2020): 1-11.
>
> Response: We thank the reviewer for the constructive comment. For the stochastic subgradient descent method, we have included a truncation step, which is novel in the context of stochastic L#-convex function minimization. We have proved that the truncation step helps reduce the computational cost by $O(d)$, which is verified by the numerical experiment. The above references did not consider the truncation step, and therefore would not achieve this reduction.
>
> 3. Theorem 6 is a simple application of Jiang (2020) via LLL algorithm (Lenstra et al., 1982).
>
> Response: We thank the reviewer for the valuable comment. We admit that the LLL algorithm used in [Jiang 2020] can reduce the number of arithmetic operations and make the algorithm more practical, but the LLL algorithm will not reduce the number of evaluations to the stochastic objective values. Furthermore, the method in [Jiang 2020] is only for deterministic problems. We have proposed a general framework to extend most deterministic cutting-plane methods to stochastic problems, whose analysis poses additional difficulties.
>
>
> 4. The lower bound result in Theorem 7 is interesting. Is there a polynomial-time algorithm for solving L# convex programs? At least we know that for [0,1} submodular minimization, it is a strongly polynomial-time solvable.
>
> Response: We thank the reviewer for the valuable comment. The multi-dimensional uniform sampling algorithm in Section 4.1 has no dependence on the scale N in the asymptotic regime when delta tends to 0. Therefore, we conjecture that there may exist a polynomial-time algorithm for solving L# convex programs (perhaps under some additional conditions), but we are not yet able to prove that rigorously.
>
>
> References:
>
> [1] Axelrod, B., Liu, Y. P., & Sidford, A. (2020). Near-optimal approximate discrete and continuous submodular function minimization. In Proceedings of the Fourteenth Annual ACM-SIAM Symposium on Discrete Algorithms (pp. 837-853). Society for Industrial and Applied Mathematics.

---

### Official Review · Reviewer_sXUy · 2021-07-15

**Rating:** 5
**Confidence:** 3

**Summary:**

This paper studies the minimization of a stochastic L$^\\natural$-convex function.
- The authors analyze the convergence of three algorithms, truncated subgradient, multi-dim uniform sampling, and stochastic cutting-plane. The time complexity of these algorithms is typically $O(d^2N^2/\\epsilon^2)$ or something higher, where $d$ is the dimension, $N$ is the range of the feasible region, and $\\epsilon$ is the error-accuracy.
- The authors also proved a lower bound that any algorithm takes $\\Omega(d/\\epsilon^2)$-time to find an $\\epsilon$-optimal solution.
- In numerical experiments, the truncated subgradient method rapidly found an $\\epsilon$-optimal solution.

**Ethics Review Area:**

["I don’t know"]

**Main Review:**

This paper has solid algorithmic results and a lower bound. The algorithms exploit recent techniques developed in fast submodular function minimization using subgradients of the Lovász extension. This paper extends these techniques towards two directions, from submodular to L$^\\natural$-convex and from exact to stochastic oracles. These extensions seem nontrivial. Therefore, the theoretical contributions are novel and significant.

On the other hand, the writing is unclear in several places. There is undefined terminology, and the table of the experiment is difficult to understand. Furthermore, the lower bound result looks weaker than the known result.
- Known sub-optimality gap: What does the known sub-optimality gap mean exactly? This concept seems crucial as it reduces $N$ to $\\log N$ in the time complexity. Unfortunately, almost nothing was explained. Is it equivalent to that the function is $1/c$-integer valued?
- Lower bound: Ito (2019) proved an $\\Omega(d/\\sqrt{T})$ lower bound for the additive error of stochastic submodular function minimization, where $T$ is the number of oracle calls. This translates to $\\Omega(d^2/\\epsilon^2)$ oracle calls, which is larger than the claimed $\\Omega(d/\\epsilon^2)$ lower bound on the time complexity. How can I compare these lower bounds?
- Experiments: Tables 2 and 3 are difficult to understand. What does "Cost" mean there? Is it the running time in sec? Furthermore, the range of $d$ and $N$ are somewhat small. Since the method used in the experiment is first-order, should it be scalable to larger instances?

For now, I would wait for the authors' response.


**Time Spent Reviewing:**

4

---

> ### Author Response · Authors · 2021-08-09
> **Response to Reviewer sXUy**
>
> We thank the reviewer for the detailed comments and suggestions. Please find our response to the comments below.
>
> 1. What does the known sub-optimality gap mean exactly? This concept seems crucial as it reduces N to log(N) in the time complexity. Unfortunately, almost nothing was explained. Is it equivalent to that the function is 1/c-integer valued?
>
> Response: We thank the reviewer for the valuable comment and we are sorry for the confusion. If the sub-optimality gap is c, then the optimal objective value is at least c smaller than the objective values of other non-optimal solutions. If the sub-optimality gap is known, then we have proved that the Lovasz extension satisfies the weak sharp minima property and acceleration techniques can be applied to reduce the dependence on N. We will provide the definition of the sub-optimality gap in the revised paper.
>
> 2. Ito (2019) proved an $\Omega(d/\sqrt{T})$ lower bound for the additive error of stochastic submodular function minimization, where T is the number of oracle calls. This translates to $\Omega(d^2/\epsilon^2)$ oracle calls, which is larger than the claimed $\Omega(d/\epsilon^2)$ lower bound on the time complexity. How can I compare these lower bounds?
>
> Response: We thank the reviewer for the very helpful comment. In [Ito et al. 2019], the goal is to minimize the *expected* objective value, while the goal of our work is to minimize the objective value *with high probability*. This difference prohibits the application of the results and the methods in [Ito et al. 2019], and the lower bound in our work is novel. We will add this comparison to the revised paper.
>
> 3. Tables 2 and 3 are difficult to understand. What does "Cost" mean there? Is it the running time in sec? Furthermore, the range of d and N are somewhat small. Since the method used in the experiment is first-order, should it be scalable to larger instances?
>
> Response: We thank the reviewer for the constructive comment. The cost refers to the average computational cost, which is defined to be the number of stochastic objective function evaluations. We will add more details to Tables 2 and 3 in the revised paper. Although the instances we have considered in this work have relatively small sizes, we can already observe the trends of the performances of the proposed algorithms, which are consistent with our theoretical findings. We believe that it is possible to implement the proposed algorithms on larger instances with more computational resources.
>
>
> In summary, we believe that we can fully address the reviewer's comments by providing more clarifications and discussions. We hope that the reviewer will kindly reconsider the rating.

---

> > ### Comment · Reviewer_sXUy · 2021-09-09
> > **On lower bound**
> >
> > I have read the other reviews and authors' comments.
> >
> > Still, I believe the lower bound is weaker than (Ito 2019). The authors proved $\Omega(d/\epsilon^2)$ lower bound, whereas Ito proved $\Omega(d^2/\epsilon^2)$. In the rebuttal comment, the authors claim their lower bound is a high probability bound and Ito is only an expectation bound.
> >
> > But *how can an expectation bound be larger than a high probability bound?* If an algorithm achieves an $\epsilon$ error w.h.p., then the expected error must be $O(\epsilon)$ (as long as the objective function is bounded by a constant). So, the high probability bound should be larger than the expectation bound.
> >
> > For the algorithmic side, the author clarified the definition of "sub-optimality gap" and "costs" in the experiment table. The authors should include them in the revised version. Now I think the algorithmic part (upper bound) are fine.
> >
> > Due to the above concern on the lower bound, I'll keep my current score for now.

---

> > > ### Author Response · Authors · 2021-09-09
> > > **Response**
> > >
> > > We would like to thank the reviewer for the valuable comment. We agree that the lower bound established in this work is a little counter-intuitive, when being compared to Ito (2019). We hope to elaborate on the comparison and provide perspectives that the lower bound in our work is not weaker than Ito (2019).
> > >
> > > 1. The lower bound $\Omega(d/\epsilon^2)log(1/\delta)$ for the high-probability case has this $\delta$-related extra term $\log(1/\delta)$, where $\delta$ represents the failing probability. Suppose one has an algorithm achieving an $\epsilon$ error with high probability $1-\delta$. If one wants to guarantee that the expected error is $\epsilon$ (with the intention that $\epsilon$ is a small enough number), the failing probability $\delta$ needs to be small enough. This choice of $\delta$ will depend on $\epsilon$ and the range of the objective function (say, we denote the gap between the maximal value and minimal value of the objective function as M). Specifically, with a high-probability $\epsilon$ error in hand, to ensure that the expected error is smaller than $\epsilon$, one needs $(1-\delta)*\epsilon + \delta * M \leq \epsilon$. This is equivalent to requiring $1/\delta >= M/\epsilon$. We hope to note that our lower bound result is established for an arbitrary L-convex function and does not rely on $M$. Therefore, one can choose $M$ to be sufficiently large, e.g., $M = e^{d^2}$. Then to ensure an $\epsilon$ expected error bound from an $\epsilon$ high-probability error bound, it is required that $1/\delta >= e^{d^2}/\epsilon$, in which case, the lower bound order $(d/\epsilon^2)log(1/\delta)$ becomes at least $d^3/\epsilon^2$. We hope to use the above arguments to explain that the high probability bound and the expectation bound can be different and it may not always be the case that a high probability bound straightforward leads to an expectation bound with the same dependence on d and $\epsilon$. The reason in summary is that the additional term $\log(1/\delta)$ in high probability bound may not be negligible given our results for arbitrary L-convex functions.
> > >
> > > 2. Inspired by the reviewer’s comment on lower bounds, we note that the high probability upper bound provided in our work is also O(d) smaller than the corresponding upper bound in Ito (2019) regarding expected error. This is analogous to the reviewer’s observation on comparing the lower bound between our work and Ito (2019), where our lower bound is O(d) smaller than the Ito’s. The O(d) difference is favoring our result on upper bound and favoring Ito (2019) on lower bound. However, this O(d) difference between our work and Ito (2019) on upper/lower bounds may need to be jointly considered with the additional $O(\log(1/\delta))$ term in the high probability bounds.
> > >
> > > In the revision, we plan to add detailed discussions on the comparison between our work and Ito (2019), enlightened by the reviewer’s comment and suggestion. We hope that our explanation could settle the reviewer's concern and the reviewer could kindly reconsider the rating.

---

### Official Review · Reviewer_QmRr · 2021-07-17

**Rating:** 6
**Confidence:** 3

**Summary:**

This paper focuses on the problem of minimizing $f(x) = \mathbb{E}_{\xi}[F(x,\xi_x)]$ where $f:[N]^d\to\mathbb{R}$ is an $L^\natural$ function. This class of functions can be interpreted as a discrete analog of convex functions. The goal is to design polynomial time algorithms that guarantee a solution whose value is $\epsilon$ away from the optimal value with probability at least $1-\delta$ (in the paper referred as $(\epsilon,\delta)$-PAC guarantee).

The authors provide upper bounds and a lower bound for the computational complexity of the problem, which is defined as the worst-case expected time over all possible $(\epsilon,\delta)$-PAC algorithms. First, the authors present the standard Lovasz extension for $L^\natural$ functions. This extension is convex so common techniques in convex optimization can be used. First, they present a truncated stochastic subgradient algorithm that has an appropriate rounding procedure as a final step. They show this method has a complexity of $\tilde{O}(\frac{d^2N^2}{\epsilon^2}\log(1/\delta))$ when $f$ is $L$-Lipschitz. Later, they show that under the assumption that the subgradients of the function are uniformly bounded with probability 1, the same algorithm without truncation has an improved complexity. The authors also study the performance of the algorithms when the suboptimality gap is known. Then, they give two methods for the case when the Lipschitz constant is not known: multidimensional uniform sampling and dimension reduction. Finally, they give an information-theoretical lower bound of $\Theta(\frac{d}{\epsilon^2}\log(1/\delta))$.


**Limitations And Societal Impact:**

The authors discuss some of the limitations. The work is purely theoretical so there are no clear potential societal impacts

**Main Review:**

The paper is overall well written and easy to follow. I think one of the main contributions of the paper is the rounding technique applied in the truncated gradient descent. The rest of the analysis uses standard techniques in convex optimization, which is possible since the Lovasz extension is convex. It would be great if the authors give more intuition on the role that the truncation plays, more than simply stating that it reduces the complexity. Also, they should mention if this technique has been used before in other contexts.

The lower bound seems a direct adaptation of the techniques used in [Zhang et al. 2021] and [Graur et al. 2020]; the authors should comment if there is any new technical contribution.

It is hard to analyze the impact of the results in Section 4, in particular because they heavily rely on two preprints [Zhang et al. 2021] and [Jiang 2020], and most of the analyses are in the Appendix.

I think the paper has its merits, but the authors should emphasize the challenges that the $L^\natural$ functions pose in each given analysis, in this way the technical contributions would be highlighted. Otherwise, my feeling (without being an expert on the specific techniques that are used) is that the paper forgets about $L^\natural$ functions once the Lovasz extension is given, since the whole toolkit of convex optimization can be used. I carefully checked some of the proofs in the Appendix and, for example, it is not clear what role the Lovasz extension plays in the proof of Theorem 3 (Appendix C.2), besides convexity.

Minor comments:
-	Page 2 Line 53: reference should be Hazan & Kale (2011)?
-	Page 2 Table 1: caption mentions $M$ but this term is not there.
-	Page 3 line 104: an
-	Page 3 line 119: algorithms -> algorithm
-	Page 4 line 137: clarify that $[1,N]^d$ corresponds to the continuous cube to avoid confusion with the discrete set $[N]^d$
-	Page 5 Alg. 2: in line 1, $T$ refers to transpose?
-	Page 7: there is a text in blue
-	Page 8 line 320: objective values
-	Appendix eq (17): extra N?


**Time Spent Reviewing:**

13

---

> ### Author Response · Authors · 2021-08-09
> **Response to Reviewer QmRr**
>
> We thank the reviewer for the detailed comments and suggestions. Please find our response to the comments below.
>
> 1. It would be great if the authors give more intuition on the role that the truncation plays, more than simply stating that it reduces the complexity. Also, they should mention if this technique has been used before in other contexts.
>
> Response: We thank the reviewer for the constructive comment. The intuition behind the truncation operation is that the truncation makes the norm of the subgradient smaller compared to no truncation, which in turn allows the choice of a larger step size. With the larger step size, the convergence rate can be improved. Using the truncation operation to accelerate the stochastic subgradient descent method is novel in the context of the stochastic $L^\natural$-convex function minimization problem. In the literature, the general idea of truncation has been considered in other contexts, such as the sparse phase retrieval problem [1]. We will include the intuition and the discussion of the truncated methods in other contexts in the revised paper.
>
> 2. The lower bound seems a direct adaptation of the techniques used in [Zhang et al. 2021] and [Graur et al. 2020]; the authors should comment if there is any new technical contribution.
>
> Response: We thank the reviewer for the valuable comment. We have borrowed some tools from [Zhang et al. 2021], which further relies on the information-theoretical inequality in [Kaufmann et al. 2016]. However, the results in [Zhang et al. 2021] are only for the one-dimensional case, while our results are for the multi-dimensional case. The construction in [Graur et al. 2020] is for submodular functions. Therefore, we have made modifications to the construction and provided an extension to $L^\natural$-convex functions. The extension is non-trivial and is a major technical contribution of this section.
>
> 3. It is hard to analyze the impact of the results in Section 4, in particular because they heavily rely on two preprints [Zhang et al. 2021] and [Jiang 2020], and most of the analyses are in the Appendix.
>
> Response: We thank the reviewer for the very helpful comment. The dimension reduction method is the first algorithm for the stochastic $L^\natural$-convex function minimization that is strongly polynomial and does not require prior knowledge about the objective function except the $L^\natural$-convexity. In addition, we admit that the LLL algorithm used in [Jiang 2020] can reduce the number of arithmetic operations and make the algorithm more practical, but the LLL algorithm will not reduce the number of evaluations to the stochastic objective values. Furthermore, the method in [Jiang 2020] is only for deterministic problems. We have proposed a general framework to extend most deterministic cutting-plane methods to stochastic problems, whose analysis is non-trivial.
>
> 4. I think the paper has its merits, but the authors should emphasize the challenges that the L-functions pose in each given analysis, in this way the technical contributions would be highlighted. Otherwise, my feeling (without being an expert on the specific techniques that are used) is that the paper forgets about L-functions once the Lovasz extension is given, since the whole toolkit of convex optimization can be used. I carefully checked some of the proofs in the Appendix and, for example, it is not clear what role the Lovasz extension plays in the proof of Theorem 3 (Appendix C.2), besides convexity.
>
> Response: We thank the reviewer for the constructive comment. Besides the convexity, our results rely heavily on other two important properties of L# convex functions. The first property is that the global minima are integral points. This property makes it possible to design strongly polynomial algorithms; please see Section 4. The second property is that the Lovasz extension is a piecewise linear function. When the sub-optimality gap is known, we have proved that the Lovasz extension satisfies the Weak Sharp Minima property and the subgradient descent method can be accelerated to achieve the $O(\log(N))$ dependence on the scale N; please see Appendix C.5.
>
>
> In summary, we believe that we can fully address the review comments by adding more clarifications and discussions, and hope that the reviewer will kindly reconsider the rating.
>
> Reference:
>
> [1] Wang, G., Zhang, L., Giannakis, G.B., Akçakaya, M. and Chen, J., 2017. Sparse phase retrieval via truncated amplitude flow. IEEE Transactions on Signal Processing, 66(2), pp.479-491.

---

> > ### Comment · Reviewer_QmRr · 2021-09-14
> > **response**
> >
> > I have read the other reviews and comments. The authors have addressed my concerns. I increase my score one point.

---

### Official Review · Reviewer_CCmo · 2021-07-17

**Rating:** 7
**Confidence:** 3

**Summary:**

In this paper, the authors studied the problem of stochastic $L^{\natural}$-convex function minimization. This problem is a extension of submodular minimization. Authors come up with an algorithm that uses projected gradient descent with truncated gradient on lovasz extension. They also provide lower bound for the complexity of this problem.

**Limitations And Societal Impact:**

The authors discussed the limitations and potential negative social impact of their work.



**Main Review:**

1)Author motivates the subject from a theoretical and practical point of view.
2)In the introduction, the author reviewed the previous works. Still, related papers like [1,2] have not been appropriately discussed.

3)the author successfully addressed the issue and recommended the algorithms.

4)The authors successfully provide simulation results for their algorithm.

5)The paper is well-organized and well-written.

6)The proof is mathematically correct.

7)the code is not included.


[1]Chakrabarty, D., Lee, Y. T., Sidford, A., & Wong, S. C. W. (2017, June). Subquadratic submodular function minimization. In Proceedings of the 49th Annual ACM SIGACT Symposium on Theory of Computing (pp. 1220-1231).

[2]Axelrod, B., Liu, Y. P., & Sidford, A. (2020). Near-optimal approximate discrete and continuous submodular function minimization. In Proceedings of the Fourteenth Annual ACM-SIAM Symposium on Discrete Algorithms (pp. 837-853). Society for Industrial and Applied Mathematics.


----------- Post Rebuttal-------------

I have read all the comments, and I don't change my score.

**Time Spent Reviewing:**

4

---

> ### Author Response · Authors · 2021-08-09
> **Response to Reviewer CCmo**
>
> We thank the reviewer for the detailed comments and suggestions. Please find our response to the comments below.
>
> 1. In the introduction, the author reviewed the previous works. Still, related papers like [1,2] have not been appropriately discussed.
>
> Response: We thank the reviewer for pointing out these references. We will add some discussions in the revised paper. Specifically, the submodular minimization problems considered in papers [1,2] are related to the problems in this work. Our problems can be viewed as stochastic versions of the problems in [1,2], which also naturally cause the difference in algorithm design and analysis. The efficiency of the algorithms in [1,2] comes from on sampling a stochastic subgradient with O(1) variance within O(1) evaluations, which is based on the preprocessing phase (Lemma 3.4 in [2]). However, the sampling phase requires an importance sampling based on the exact function values difference that is computed in the preprocessing phase. It is challenging to extend the importance sampling to the case when we only have access to noisy function values. The major techniques in our gradient-based algorithm is the truncation step, which is proved to reduce the sample complexity by O(d).
>
> 2. The code is not included.
>
> Response: We will provide the code after the revision process.

---

### Decision · Program_Chairs · 2021-09-27

**Decision:**

Accept (Poster)

**Comment:**

From the SAC: after looking at the paper, reviews, rebuttal, and discussion, and questions about reviewer sXUy , I am going to concur with the AC's recommendation of accept as a poster. However, this was a borderline paper compared to other papers, so I strongly urge you to take these reviewer comments into account. By your own words, "We agree that the lower bound established in this work is a little counter-intuitive, when being compared to Ito (2019)", therefore, it behooves you to make sure in the next version of this paper that this fact is acknowledged in the paper, and that you strive in your edits to make it as non-counter-intuitive as possible. Also, please ensure that all questions and concerns raised by the reviewer are addressed in the next version of the paper.